# MOSTLY EXPLORATION-FREE ALGORITHMS FOR MULTI-OBJECTIVE LINEAR BANDITS

## ABSTRACT

We address the challenge of solving multi-objective bandit problems, which are increasingly relevant in real-world applications where multiple possibly conflicting objectives must be optimized simultaneously. Existing multi-objective algorithms often rely on complex, computationally intensive methods, making them impractical for real-world use. In this paper, we propose a novel perspective by showing that objective diversity can naturally induce free exploration, allowing for simpler, near-greedy algorithms to achieve optimal regret bounds up to logarithmic factors with respect to the number of rounds. We introduce simple and efficient algorithms for multi-objective linear bandits, which do not require constructing empirical Pareto fronts and achieve a regret bound of $\widetilde{\mathcal{O}}(\sqrt{T})$ under sufficient objective diversity and suitable regularity. We also introduce the concept of objective fairness, ensuring equal treatment of all objectives, and show that our algorithms satisfy this criterion. Numerical experiments validate our theoretical findings, demonstrating that objective diversity can enhance algorithm performance while simplifying the solution process.

## 1 INTRODUCTION

Multi-objective decision-making problems have become increasingly prevalent in today's complex, real-world applications. From recommendation systems to robotics, decision-makers often face the need to optimize multiple conflicting objectives simultaneously. This complexity naturally leads to the formulation of *multi-objective bandit problems* (Drugan & Nowe, 2013; Turgay et al., 2018; Lu et al., 2019; Xu & Klabjan, 2023; Cheng et al., 2024; crepon et al., 2024; Kim et al., 2023), a generalization of the single-objective bandit framework. However, solving multi-objective bandit problems is particularly challenging due to the added complexity of balancing the exploration-exploitation tradeoff across multiple objectives.

To address this challenge, many multi-objective bandit algorithms often resort to complex, sometimes computationally intractable methods (Yahyaa & Manderick, 2015; Turgay et al., 2018; Lu et al., 2019; Kim et al., 2023), especially when compared to their simpler single-objective counterparts (Abbasi-Yadkori et al., 2011; Chu et al., 2011; Chapelle & Li, 2011; Agrawal & Goyal, 2013; Abeille & Lazaric, 2017). Many of these methods often involve constructing empirical Pareto fronts in each round, leading to significant computational overhead, making them less practical for real-world deployment.

Despite the additional complexity that multiple objectives introduce, we aim to address the following intriguing research question:

*Can the presence of multiple diverse objectives actually facilitate learning rather than hinder it?*

To our best knowledge, this question has never been addressed. To some, it may appear even somewhat counter-intuitive since a larger number of objectives usually implies a more challenging problem setup. Yet, we ask whether the diversity in objectives can induce exploration, enabling simpler near-greedy algorithms to achieve performance guarantees of statistical efficiency which is typically obtained by more complex approaches.

In this work, we propose a novel perspective on this problem by showing that *objective diversity*—rather than feature diversity, which is commonly assumed in the literature—can drive exploration in multi-objective linear bandits. This new insight allows us to demonstrate the statistical efficiency of simple round-robin greedy algorithms, such as the algorithms we introduce, `MORR-Greedy` and `MORO-Greedy`. Remarkably, our algorithms achieve optimal regret bounds with respect to the number of rounds $T$ without the need for the context (feature) diversity assumption that underpins most of the existing greedy bandit literature (Kannan et al., 2018; Raghavan et al., 2018; Hao et al., 2020; Bastani et al., 2021).

While most previous works on greedy bandits (Kannan et al., 2018; Raghavan et al., 2018; Bastani et al., 2021) rely on context diversity to induce exploration, we *do not assume any such diversity in the features*. Our algorithms even perform effectively in fixed feature settings, where previous greedy approaches and their proof techniques fail. To our best knowledge, this finding represents the first result where a greedy-type algorithm achieves no regret results without relying on feature diversity in parametric bandits. Furthermore, this work is the first to study the provable efficiency of simple round-robin algorithms for multi-objective linear bandits.

We rigorously analyze our algorithms in terms of both regret performance and objective fairness, demonstrating their theoretical and empirical superiority under objective diversity and suitable regularity, even without feature diversity. Our work introduces a new perspective in multi-objective bandit research, showing that, in certain cases, *more and diverse objectives can help rather than complicate the learning process*.

Our main contributions are summarized as follows.

## 1.1 CONTRIBUTIONS

- We rigorously analyze a new and sufficient condition for the multi-objective bandit problem, under which near-greedy algorithms achieve statistical efficiency without relying on the commonly assumed context diversity condition in the greedy bandit literature (Kannan et al., 2018; Raghavan et al., 2018; Hao et al., 2020; Bastani et al., 2021). This result is driven by the free exploration enabled by the diversity of objectives. Notably, this free exploration occurs even in fixed context settings, not just stochastic environments. Our key insight is that having multiple objectives simplifies rather than complicates the problem, enhancing the performance of the algorithms.

- We propose two simple yet highly efficient algorithms, `MORR-Greedy` and `MORO-Greedy`, for multi-objective linear bandits. Unlike many existing multi-objective algorithms, these algorithms do not require constructing an empirical Pareto optimal set of arms in each round, resulting in simpler implementation and lower computational overhead.

- We establish that `MORR-Greedy` and `MORO-Greedy` are statistically efficient under objective diversity and regularity assumptions, achieving a regret bound of $\widetilde{\mathcal{O}}(\frac{\sqrt{dT}}{\lambda_0})$, where $d$ is the dimension of the feature vectors, $T$ is the total number of rounds, and $\lambda_0$ is the diversity index of objective parameters defined in Section 4.

- We introduce a novel evaluation criterion for multi-objective bandit algorithms, called *objective fairness*, which ensures that all objectives are treated equitably, with no objective being neglected. We rigorously prove that both `MORR-Greedy` and `MORO-Greedy` satisfy this principle.

- Through extensive numerical experiments, we demonstrate that `MORR-Greedy` consistently outperforms existing multi-objective methods across a wide range of scenarios. These results empirically validate our theoretical claims, showing that the diversity of objectives not only facilitates exploration but also significantly enhances algorithmic performance.

## 1.2 RELATED WORK

The multi-objective bandit problem, an extension of the single-objective bandit framework that captures real-world scenarios with multiple conflicting optimization objectives, was first introduced by Drugan & Nowe (2013). They proposed two approaches using the UCB algorithm: one based on Pareto optimality and the other on scalarization. The scalarization approach simplifies the problem by converting it into a single-objective one, using weighted combinations of objectives. In contrast,

Table 1: Comparison with Related Work. $K$ is the total number of arms, $d$ is the dimension of feature vectors, $T$ is the time horizon, and $\Delta$ denotes the minimum Pareto regret over suboptimal arms.

| Paper | Objective | Parametric | Free Exploration | Regret |
|---|---|---|---|---|
| Drugan & Nowe (2013) | Multiple | $\times$ | $\times$ | $\mathcal{O}(\frac{K}{\Delta}\log T)$ |
| Lu et al. (2019) | Multiple | $\bigcirc$ | $\times$ | $\widetilde{\mathcal{O}}(d\sqrt{T})$ |
| Cheng et al. (2024) | Multiple | $\bigcirc$ | $\times$ | $\widetilde{\mathcal{O}}((dT)^{2/3})$ |
| Kim et al. (2023) | Multiple | $\bigcirc$ | $\times$ | $\mathcal{O}(d^3\log dT + \frac{d}{\Delta}\log\frac{dT}{\Delta})$ |
| Kannan et al. (2018) | Single | $\bigcirc$ | $\bigcirc^\dagger$ | $\widetilde{\mathcal{O}}(\frac{\sqrt{dT}}{\sigma^2})$ |
| Bastani et al. (2021) | Single | $\bigcirc$ | $\bigcirc^\dagger$(with margin cond.) | $\mathcal{O}(dK\log T)$ |
| Bayati et al. (2020) | Single | $\times$ | $\bigcirc$ | $\widetilde{\mathcal{O}}(TK^{-1/3} + K)\ddagger$ |
| **This work** | Multiple | $\bigcirc$ | $\bigcirc$ | $\widetilde{\mathcal{O}}(\sqrt{dT})$ |

$\dagger$ With the diversity assumption on context distribution. In Kannan et al. (2018), $\sigma^2$ refers to the standard deviation of Gaussian perturbation applied to the contexts.
$\ddagger$ Bayesian regret, which is a weaker notion of regret compared to frequentist (worst-case) regret.

the Pareto optimality approach treats all objectives equally, without making any assumptions about their relationships. This second approach inspired numerous studies on multi-objective bandits focused on Pareto efficiency (Turgay et al., 2018; Lu et al., 2019; Xu & Klabjan, 2023; Cheng et al., 2024; crepon et al., 2024; Kim et al., 2023).

Recent advancements have extended the multi-objective bandit framework to linear contextual settings. Lu et al. (2019) established theoretical regret bounds for the UCB algorithm within the generalized linear bandit framework. Cheng et al. (2024) introduced two algorithms for multi-objective stochastic linear bandits under a hierarchy-based Pareto dominance condition. These methods differ based on how objectives are structured hierarchically, with their regret bounds compared in Table 1. Additionally, Kim et al. (2023) explored Pareto front identification in linear bandit settings, focusing on algorithms that prioritize learning the optimal Pareto set.

While these works made important strides, they largely overlook the potential for free exploration that can arise from the diversity of objectives, particularly in the absence of context diversity. Recent research on single-objective linear contextual bandits with stochastic contexts has shown that if context diversity is sufficiently high, greedy algorithms can achieve near-optimal regret bounds (Bastani et al., 2021; Kannan et al., 2018; Raghavan et al., 2018; Hao et al., 2020). However, the extension of these results to multi-objective bandits has been limited by an assumption of context diversity, leaving a gap in understanding how exploration can occur without it.

Our work addresses this gap by focusing on free exploration driven by objective diversity, even in the absence of context diversity. While Bayati et al. (2020) demonstrated that greedy algorithms perform well in non-contextual single-objective settings when the number of arms is large, they relied on a $\beta$-regularity assumption related to the reward distribution. In contrast, we introduce the concept of $\gamma$-regularity (Definition 6), which extends the notion of regularity to feature spaces in the multi-objective setting. Unlike previous work, which provided only Bayesian regret bounds, we rigorously establish worst-case regret bounds for our proposed algorithms, `MORR-Greedy` and `MORO-Greedy`, under this new regularity assumption.

Our research also contributes by showing that free exploration can occur even in fixed context settings, driven solely by the diversity of the objectives. This is the first time that a theoretical guarantee has been provided for such exploration in multi-objective linear bandits, without relying on context diversity—a significant departure from existing literature.

## 2 PROBLEM SETTINGS

### 2.1 NOTATIONS

We denote by $[n]$ the set $\{1, \ldots, n\}$ for a positive integer $n$. We use $||x||_2$ to denote the $l_2$ norm of vector $x \in \mathbb{R}^d$ and $||x||_A = \sqrt{x^\top A x}$ to denote the weighted norm of $x$ induced by a positive definite matrix $A \in \mathbb{R}^{d \times d}$. We write $\mathbb{B}_R^d = \{x \in \mathbb{R}^d \mid ||x||_2 \leq R\}$ and $\mathbb{S}_R^{d-1} = \{x \in \mathbb{R}^d \mid ||x||_2 = R\}$. When $d$ is clear in the context, we just use $\mathbb{B}_R := \mathbb{B}_R^d$ and $\mathbb{S}_R := \{x \in \mathbb{R}^d \mid ||x||_2 = R\}$. Finally, $\mathbf{1}_{\text{condition}}$ means the identity function having value 1 if the condition is true, and 0 otherwise .

### 2.2 MULTI-OBJECTIVE LINEAR BANDITS

In each round $t = 1, 2, \ldots, T$, each feature vector $x_i \in \mathbb{R}^d$ for $i \in [K]$ is associated with stochastic reward $y_{i,m}(t)$ for objective $m \in [M]$ with mean $x_i^\top \theta_m^*$ where $\theta_m^* \in \mathbb{R}^d$ is a fixed, unknown parameter. While we present our problem setting in the fixed feature setup for clear exposition of our main idea, we also present our results under varying context setting in Appendix B. After the agent pulls an arm $a(t) \in [K]$, the agent receives a stochastic reward vector $y_{a(t)}(t) = \left(y_{a(t),1}(t), \ldots, y_{a(t),M}(t)\right) \in \mathbb{R}^M$ as a bandit feedback, where $y_{a(t),m}(t) = x_{a(t)}^\top \theta_m^* + \eta_{a(t),m}(t)$ and $\eta_{a(t),m}(t) \in \mathbb{R}$ is a zero mean noise for objective $m \in [M]$. To simplify notation, we denote by $x(t) := x_{a(t)}$ and $y(t) := y_{a(t)}(t)$ with slight notational overloading the selected arm vector in round $t$ and its rewards respectively. We assume that for all $m \in [M]$, $\eta_{a(t),m}(t)$ is conditionally $\sigma^2$-sub-Gaussian with $\sigma > 0$, i.e., for all $\lambda \in \mathbb{R}$, $\mathbb{E}[e^{\lambda \eta_{a(t),m}(t)} | \mathcal{F}_{t-1}] \leq \exp\left(\lambda^2 \sigma^2 / 2\right)$ where $\mathcal{H}_t$ is the history $\left(\{x(s)\}_{s \in [t]}, \{a(s)\}_{s \in [t]}, \{y(s)\}_{s \in [t]}\right)$ and $\mathcal{F}_t$ is the $\sigma$-algebra generated by $\mathcal{H}_t$ and $x(t+1)$.

#### 2.2.1 PARETO REGRET METRIC

In this work, we use the notion of Pareto regret (Drugan & Nowe, 2013; Turgay et al., 2018; Lu et al., 2019; Xu & Klabjan, 2023; Cheng et al., 2024; crepon et al., 2024; Kim et al., 2023) as the performance metric for the multi-objective bandit algorithms. Before we formally define the Pareto regret, we first introduce the notions of Pareto order and Pareto front.

**Definition 1** (Pareto Order). *For $u = \left(u_1, \ldots, u_M\right)$, $v = \left(v_1, \ldots, v_M\right) \in \mathbb{R}^M$, the vector $u$ dominates $v$, denoted by $v \prec u$, if and only if $v_m \leq u_m$ for all $m \in [M]$, and there exists $m' \in [M]$ such that $v_{m'} < u_{m'}$. We use notations $v \not\prec u$ when $v$ is not dominated by $u$, and $u \parallel v$ when $u$ and $v$ are incomparable, i.e., either $u$ or $v$ are not dominated by the other, respectively.*

**Definition 2** (Pareto Front). *Let $\mu_i \in \mathbb{R}^m$ be the expected reward vector of arm $i \in [K]$. Then, arm $i$ is Pareto optimal if and only if $\mu_i$ is not dominated by $\mu_{i'}$, for all $i' \in [K]$. The Pareto front is the set of all Pareto optimal arms.*

**Definition 3** (Pareto Regret). *We denote **Pareto suboptimality gap** $\Delta_i$ for arm $i \in [K]$ as the infimum of the scalar $\epsilon \leq 0$ such that $x_i$ becomes Pareto optimal after adding $\epsilon$ to all entries of its expected reward. Formally,*

$$\Delta_i := \inf\left\{\epsilon \mid (\mu_i + \epsilon) \not\prec \mu_{i'}, \forall i' \in [K]\right\}.$$

*Then, the cumulative **Pareto regret** is defined as $\mathcal{PR}(T) := \sum_{t=1}^T \mathbb{E}[\Delta_{a(t)}]$, the cumulative Pareto suboptimality gap $\Delta_{a(t)}$ of the arms pulled by the learner.*

The goal of the agent is to minimize the cumulative Pareto regret while ensuring fairness over objectives, which is described in the next section.

#### 2.2.2 OBJECTIVE FAIRNESS

Pareto regret minimization is a central goal in multi-objective bandit algorithms, but it does not fully capture the multi-objective nature of the problem. Focusing solely on Pareto regret minimization allows algorithms to optimize for a single specific objective, potentially neglecting others, while achieving regret bounds comparable to those in single-objective settings (Xu & Klabjan, 2023). Therefore, meaningful multi-objective bandit algorithms should aim to balance multiple objectives,

typically incorporating additional considerations such as fairness, alongside Pareto regret minimization.

In the context of multi-objective bandits, fairness reflects the algorithm's impartial treatment of multiple equivalent objectives. The earliest notion of fairness, to the best of our knowledge, was introduced by Drugan & Nowe (2013) and emphasizes equal representation across all Pareto fronts. This concept is closely linked to Pareto front approximation and remains a key objective for many existing algorithms (Drugan & Nowe, 2013; Yahyaa & Manderick, 2015; Turgay et al., 2018; Lu et al., 2019). However, in practice, the true Pareto front is often unknown, requiring most Pareto-efficient algorithms to rely on empirical Pareto front approximations during the selection process. This reliance introduces two significant challenges: the lack of theoretical guarantees concerning the fairness of true Pareto optimal arms, and the computational overhead associated with constructing the empirical Pareto front in each round.

To address these limitations, we propose a new notion of fairness in multi-objective bandit problems, focusing on the *fairness of objectives*. Specifically, we advocate for algorithms that consistently consider all optimal arms for each objective, ensuring that no objective is neglected over time. This perspective shifts the focus from approximating the Pareto front to ensuring that each objective is adequately and equitably addressed throughout the decision-making process.

**Definition 4** (Objective fairness)**.** *Let $\mu_{i,m}$ be the expect reward of arm $i$ for objective $m$, $a_m^*$ be the arm that has the largest reward for objective $m$, and $\mu_m^* := \mu_{a_m^*,m}$. For all $\epsilon > 0$, define **the objective fairness index** $p_{\epsilon,T}$ of an algorithm as*

$$p_{\epsilon,T} := \min_{m \in [M]} \left( \frac{1}{T} \mathbb{E} \left[ \sum_{t=1}^{T} \mathbf{1}_{\{\mu_m^* - \mu_{a(t),m} < \epsilon\}} \right] \right).$$

*Then, we say that the algorithm satisfies the **objective fairness** if there exists a positive lower bound $B$ that satisfies the following conditions:*
*1. $\lim_{T \to \infty} p_{\epsilon,T} \geq B$,*
*2. $B$ does not include a term with the number of arms $K$.*

The objective fairness index measures the proportion of rounds in which the $\epsilon$-optimal arms are selected for the least chosen objective. This index provides a means to evaluate how fairly the algorithm treats the near-optimal arms of each objective. If $p_{\epsilon,T} \approx \frac{1}{M}$, the algorithm is almost perfectly fair to each objective, while $p_{\epsilon,T} \approx 0$ indicates that the algorithm neglects at least one objective. Intuitively, objective fairness is an asymptotic concept that ensures the proportion of selecting near-optimal arms remains balanced across all objectives over time. Condition 2 is included to impose a constraint that ensures the strategy performs better than a random strategy, which selects all arms with equal probability.

**Remark 1.** *Most existing multi-objective algorithms constructing the empirical Pareto front for each rounds are unlikely to satisfy the objective fairness criterion, because the empirical Pareto front continuously changes over time $t$.*

## 3 PROPOSED ALGORITHM

### 3.1 MULTI-OBJECTIVE ROUND ROBIN – GREEDY ALGORITHM

We propose a new algorithm named the `MORR-Greedy` algorithm, which selects arms greedily in a round-robin manner for each objective. At first, the algorithm greedily selects arms based on the initial parameters $\beta_1, \ldots, \beta_M$, until the minimum eigen value of the Gram matrix $V_{t-1} = \sum_{s=1}^{t-1} x(s)x(s)^\top$ exceeds a certain threshold $\lambda$. After the initial rounds, the algorithm selects arms greedily, using the OLS estimators $\hat{\theta}_m(t)$ of $\theta_m^*$ as targets, iteratively. We describe our algorithm targets each objective once per round to make analysis simpler, however, when the importance of the objectives varies, we can adjust the frequency of each objective accordingly.

Most of existing algorithms regarding Pareto efficiency construct the empirical Pareto front on each round, resulting in complex algorithm structure and less practicality. Compare to other multi-objective bandit algorithms, our proposed algorithm is very easy to implement and has significantly lower computational overhead. Aside from these advantages, surprisingly, our simple algorithm can

---

**Algorithm 1** Multi-Objective Round Robin – Greedy Algorithm (`MORR-Greedy`)

---

**Require:** $T, \lambda$ {Parameters: Total rounds $T$, minimum eigenvalue threshold $\lambda$}

1: Initialize $V_0 \leftarrow 0 \times I_d$, $\beta_1, \ldots, \beta_M \in \mathbb{R}^d$
2: **for** $t = 1$ **to** $T$ **do**
3:     $m \leftarrow t \% M$ {If $m == 0$, then $m \leftarrow M$}
4:     **if** $\lambda_{\min}(V_{t-1}) > \lambda$ **then**
5:         Update the OLS estimators $\hat{\theta}_1(t), \ldots, \hat{\theta}_M(t)$
6:         Select action $a(t) \in \arg\max_{i \in [K]} x_i^\top \hat{\theta}_m(t)$
7:     **else**
8:         Select action $a(t) \in \arg\max_{i \in [K]} x_i^\top \beta_m$
9:     **end if**
10:    Observe the reward vector $y(t) = \big(y_{a(t),1}(t), \ldots, y_{a(t),M}(t)\big)$
11:    Update $V_t \leftarrow V_{t-1} + x(t)x(t)^\top$
12: **end for**

---

achieve theoretical performance guarantees (under suitable regularity) which is typically obtained by more complex algorithms.

We also introduce another multi-objective near-greedy algorithm `MORO-Greedy` (Algorithm D.1) in the appendix, which is a version of the `MORR-Greedy` algorithm that incorporates stochastic selection process and we analyze this algorithm in Appendix D.

## 3.2 Free Exploration induced by Objective Diversity

The `MORR-Greedy` algorithm (Algorithm 1) is built on the insight that exploration can arise naturally, even when the algorithm is focused solely on exploitation, provided the bandit problem involves sufficiently diverse objectives. In most of the existing multi-objective bandit literature, increasing the number of objectives complicates the problem setup and leads to more complex algorithms, especially compared to single-objective bandits.

However, we observe a surprising and beneficial side effect: the diversity of objectives can induce free exploration, enabling simple near-greedy algorithms like `MORR-Greedy` to achieve statistically efficient performance (see Theorem 1).

The core idea is that, for each objective, rounds in which greedy selections are made for other objectives can simultaneously serve as exploration rounds for the remaining objectives. In the round-robin process, exploitation occurs for one objective, while the other objectives naturally benefit from exploration. This dynamic allows for automatic exploration without incurring additional Pareto regret, providing a significant performance advantage.

This phenomenon is intuitive, yet it has not been rigorously examined in multi-objective settings until now. Our work is the first to formalize the conditions under which natural exploration can occur in the presence of objective diversity, paving the way for simpler, more efficient algorithms in multi-objective bandit problems.

## 3.3 A Strategy for Selecting Initial Parameters

To expedite the initial exploration phase, a practical strategy involves constructing a set of $M$ feature vectors that are as diverse as possible. This diversity helps ensure that each objective is sufficiently represented from the outset, enabling the algorithm to gather meaningful information early on. The following definition formalizes the properties of the initial values used in this strategy, ensuring robust exploration across all objectives.

**Definition 5** (Exploration Facilitating Initial Parameters). *For $m \in [M]$, let $z_m$ be the greedy selection among $x_1, \ldots, x_K$ for the initial objective parameter $\beta_m$. We say that initial objective parameters are exploration facilitating when a set of initial vectors $\{\beta_1, \ldots, \beta_M\}$ maximizes $\lambda_{\min}\left(\sum_{m=1}^M z_m(z_m)^\top\right)$.*

## 4 ANALYSIS

In this section, we analyze the `MORR-Greedy` algorithm from the perspectives of regret and objective fairness. Our analysis is established in the fixed feature setup to expose our main idea clearly, however we also present the similar results under the stochastic environment in Appendix B. We start with the bounded assumption similar to those used in the linear bandit literature (Abbasi-Yadkori et al., 2011; Chu et al., 2011; Agrawal & Goyal, 2013; Abeille & Lazaric, 2017).

**Assumption 1** (Boundedness). $\forall i \in [K]$, $||x_i||_2 \leq 1$, and $\forall m \in [M]$, $||\theta_m^*||_2 = 1$.

Assumption 1 is used to make a clean analysis for convenience and the first part of it is in fact standard in bandit literature. Notably, we can obtain a regret bound of the proposed algorithm that differs by a constant factor with $||x_i||_2 \leq x_{\max}$ and $l \leq \theta_m^* \leq L$ for all $m \in [M]$. We will later discuss how to extend our analysis to an arbitrary bound for feature vectors and objective parameters in Appendix E.

As stated earlier in Introduction, we are interested in the problem setting where diverse objectives play a positive role, rather than incurring hindrance. We start with simple condition that objective parameters span $\mathbb{R}^d$.

**Assumption 2** (Objective diversity). *We assume $\theta_1^*, \ldots, \theta_M^*$ span $\mathbb{R}^d$.*

In the following analysis, we define $\lambda_0 := \lambda_{\min}(\frac{1}{M} \sum_{m=1}^{M} \theta_m^* (\theta_m^*)^\top)$, which has a positive value under Assumption 2. It is important to note that we can actually relax Assumption 2 so that $\theta_1^*, \ldots, \theta_M^*$ span the spanning space of feature vectors, $span(\{x_1, \ldots, x_K\})$ (see details in Section F). That is, it can be sufficient to assume that $\theta_1^*, \ldots, \theta_M^*$ span a strict subspace of $\mathbb{R}^d$. Yet, for clear exposition of our main idea, we work with Assumption 2.

Next, we introduce the $\gamma$-regularity condition that describes the regularity on feature space in multi-objective linear bandits. The similar notion of regularity, called $\beta$-regularity, in the non-contextual MAB setup is introduced by Bayati et al. (2020). They assume the prior distribution $\Gamma$ of the expected reward $\mu$ of each arm satisfies $\mathbb{P}_\mu[\mu > 1 - \epsilon] = \Theta(\epsilon^\beta)$ for all $\epsilon > 0$. The $\gamma$-regularity extends the $\beta$-regularity to linear reward bandit problems with multiple objectives.

**Definition 6** ($\gamma$-regular condition). *For fixed $\gamma \in (0, 1]$, we say that the set of feature vectors $\{x_1, \ldots, x_K\}$ satisfies $\gamma$-regular condition if there exists $\alpha > 0$ that satisfies*

$$\forall \beta \in \mathbb{B}_\alpha(\theta_1^*) \cup \ldots \cup \mathbb{B}_\alpha(\theta_M^*), \quad \exists i \in [K], \quad x_i^\top \frac{\beta}{||\beta||_2} \geq \gamma. \tag{1}$$

We will generalize the $\gamma$-regular condition under varying context setup later in Definition 7 in Appendix B. In this case, $\gamma$-regularity condition requires the positive probability of the existence of near-optimal arms for all directions in $\mathbb{R}^d$.

**Assumption 3** ($\gamma$-regularity). *We assume $\{x_1, \ldots, x_K\}$ satisfies $\gamma_0$-regular with $\gamma_0 > 1 - \frac{\lambda_0^2}{18}$.*

Assumption 3 says that there exists at least one near optimal arm for directions in the neighborhoods of objective parameters. We can relax the existence of near optimal arms in Assumption 3 to the positive probability of existence of near optimal arms in Assumption B.1 under stochastic context setting. In comparing Assumption B.1 where $d = M = 1$ with $\beta$-regularity, we observe that $\gamma$-regularity can be viewed as a weaker notion than $\beta$-regularity. Detailed analysis on both assumptions can be found in Appendix C.2.

It is worthy to note that above assumptions are irrelevant to context diversity assumption which is commonly used in the existing greedy bandit literature (Kannan et al., 2018; Raghavan et al., 2018; Hao et al., 2020; Bastani et al., 2021). Especially, we explain the cases where $\gamma$-regularity holds but context diversity does not in Appendix C.3.

Before we start our analysis, we denote by $\alpha_0$ the value of $\alpha$ that holds the condition (1) with $\gamma_0$. If $\alpha_0$ is greater than $\psi(\lambda_0, \gamma_0) := \sqrt{\frac{\lambda_0^2}{9} - \frac{\lambda_0^4}{324}} \gamma_0 - \left(1 - \frac{\lambda_0^2}{18}\right) \sqrt{1 - \gamma_0^2}$, then we replace the value of $\alpha_0$ with $\psi(\lambda_0, \gamma_0)$. Since the condition becomes tighter as $\alpha$ increases, the $\gamma_0$-condition still holds even if the value of $\alpha_0$ is replaced by a smaller value.

## 4.1 THE REGRET BOUND OF `MORR-Greedy`

We establish the lower bound of the minimum eigenvalue of Gram matrix that increases linearly with respect to $t$. Typically in many greedy bandit approaches, the linear growth of minimum eigenvalue of Gram matrix is derived by showing a constant lower bound on $\lambda_{\min}\left(\mathbb{E}[x(t)x(t)^\top | \mathcal{H}_{t-1}]\right)$ for each round $t$ through context diversity. However, instead of leveraging context diversity, we use the diversity of the objectives to establish a constant lower bound for $\lambda_{\min}\left(\sum_{s=t_0}^{t_0+M-1} x(s)x(s)^\top\right)$ for a single cycle $s = t_0,\ t_0+1,\ \ldots,\ t_0+M-1$ in round-robin process. Let $T_0$ denote the number of rounds required until condition $\lambda_{\min}(V_t) \geq \lambda$ is satisfied.

**Lemma 1** (Minimum eigenvalue growth). *Suppose Assumptions 1, 2 and, 3 hold, and fix $\delta > 0$. If we run Algorithm 1 with $\lambda = \min\left[\frac{\sigma}{\alpha_0}\sqrt{2dT\log(\frac{dT}{\delta})},\ \frac{4\sigma^2}{\alpha_0^2}\left(\frac{d}{2}\log\left(1+\frac{2T}{d}\right)+\log\left(\frac{1}{\delta}\right)\right)\right]$, then with probability $1-2M\delta$, the following holds for the minimum eigenvalue of the gram matrix*

$$\lambda_{\min}\left(\sum_{s=1}^{t-1} x(s)x(s)^\top\right) \geq \lambda + C_0(t-T_0-M),$$

*for $T_0 + M \leq t \leq T$, where $C_0 = \lambda_0 - 2\sqrt{2 + 2\alpha_0\sqrt{1-\gamma_0^2} - 2\gamma_0\sqrt{1-\alpha_0^2}}$.*

The proof of the lemma is given in Appendix A.1.

**Remark 2.** *We can always get $C_0 \geq \frac{\lambda_0}{3}$ by setting the value of $\alpha_0$ no greater than $\psi(\lambda_0, \gamma_0) := \sqrt{\frac{\lambda_0^2}{9} - \frac{\lambda_0^4}{324}}\,\gamma_0 - \left(1 - \frac{\lambda_0^2}{18}\right)\sqrt{1-\gamma_0^2}$. In other words, this replacing process serves to increase the minimum eigenvalue of the Gram matrix at a rate $O(\lambda_0)$.*

It is well known that the minimum eigenvalue of the gram matrix increases proportionally with $t$, we can easily obtain an order of $\sqrt{T}$ regret bound. The following theorem demonstrates that the `MORR-Greedy` algorithm possesses near optimal regret.

**Theorem 1** (Pareto Regret of `MORR-Greedy`). *Suppose Assumptions 1, 2 and, 3 hold. If we run Algorithm 1 with $\lambda = \min\left[\frac{\sigma}{\alpha_0}\sqrt{2dT\log(dT^2)},\ \frac{4\sigma^2}{\alpha_0^2}\left(\frac{d}{2}\log\left(1+\frac{2T}{d}\right)+\log(T)\right)\right]$, then the Pareto regret of Algorithm 1 is bounded by*

$$\mathcal{PR}(T) \leq C_1\sqrt{2dT\log(dT)} + 4T_0 + 10M,$$

*where $C_1 = \frac{8\sigma}{\lambda_0 - 2\sqrt{2+2\alpha_0\sqrt{1-\gamma_0^2} - 2\gamma_0\sqrt{1-\alpha_0^2}}}$.*

The proof of the theorem is given in Appendix A.2.

**Discussion of Theorem 1.** The theorem demonstrates that the cumulative Pareto regret bound of `MORR-Greedy` is $\widetilde{\mathcal{O}}(\frac{\sqrt{dT}}{\lambda_0})$. Theorem 1 provides the theoretical foundation that if multiple objectives possess diversity and suitable regularity, simple round-robin type algorithms can outperform even more complicated exploration-based algorithms for multi-objective linear bandits (such phenomenon is witnessed in the experiments in Section 5).

**Remark 3.** *If the $m$ feature vectors selected greedily by the initial objective parameters that are spanning $\mathbb{R}^d$, then the minimum eigenvalue of the Gram matrix will increase proportionally with $t$ during the exploration process. In other words, when we use $\lambda$ in Theorem 1, $T_0$ can be bounded at a scale of $\widetilde{\mathcal{O}}(\min(d\log T, \sqrt{dT}))$ as long as the algorithm selects $M$ feature vectors that are spanning $\mathbb{R}^d$ during the initial Round Robin process. (In the case of fixed arms, we can always ensure this).*

The following is an argument regarding how quickly exploration can be completed. It is generally challenging to specifically determine the bound on $T_0$. However, in the `MORR-Greedy` algorithm, by using exploration facilitating initial objective parameters $\beta_1, \ldots, \beta_M$, we can get the worst-case theoretical bound on $T_0$.

**Corollary 1** (Number of Initial rounds). *Suppose Assumptions 1, 2 and, 3 hold. If we run Algorithm 1 with exploration facilitating initial objective parameters, $T_0$ can be bounded by $T_0 \leq \lfloor\frac{\lambda}{C_0}\rfloor + M$ where $C_0 = \lambda_0 - 2\sqrt{2 + 2\alpha_0\sqrt{1-\gamma_0^2} - 2\gamma_0\sqrt{1-\alpha_0^2}}$.*

The proof of the corollary is given in Appendix A.4.

## 4.2 Objective Fairness of MORR-Greedy

We confirmed that the MORR-Greedy algorithm satisfies the objective fairness. In the MORR greedy algorithm, we can obtain $l_2$ bounds for the difference between the estimators of each objective parameter and the true objective parameters. This implies that for a given $\epsilon$, with high probability, there exists $T_\epsilon$ such that we can select only near-optimal arms with a reward within an $\epsilon$ radius of the optimal reward after round $T_\epsilon$. The following theorem shows the lower bound on the ratio of selecting the near optimal arms for each objective.

**Theorem 2** (Objective Fairness of MORR-Greedy). *Given Assumptions 1 to 3, the Algorithm 1 satisfies for all $m \in [M]$,*

$$\frac{1}{T}\mathbb{E}\left[\sum_{t=1}^{T}\mathbf{1}_{\{\mu_m^* - \mu_{a(t),m} < \epsilon\}}\right] \geq \left(\frac{T - T_\epsilon - M}{MT}\right)\left(1 - \frac{3M}{T}\right),$$

*where $T_\epsilon = \max(\lfloor \frac{32\sigma^2 d \log(dT)}{(\lambda_0 - 2\sqrt{2 + 2\alpha_0\sqrt{1-\gamma_0^2} - 2\gamma_0\sqrt{1-\alpha_0^2}})^2}\frac{1}{\epsilon^2}\rfloor + T_0 + M, \ 2T_0 + 2M)$ in the same setting as Theorem 1.*

The proof of the theorem is given in Appendix A.3.

**Discussion of Theorem 2.** The theorem demonstrates that we have a lower bound on the expected proportion of selecting near optimal arms with respect to each objective by $p_{\epsilon,T} \geq \left(\frac{T - T_\epsilon - M}{MT}\right)\left(1 - \frac{3M}{T}\right)$. It is notable that $\lim_{T\to\infty}\left(\frac{T - T_\epsilon - M}{MT}\right)\left(1 - \frac{3M}{T}\right) = \frac{1}{M}$ and the limit does not include a term with $K$. This implies that our algorithm satisfies objective fairness and selects near-optimal arms for each objective equally at a ratio of $\frac{1}{M}$ as time grows. Moreover, we prove that with high probability, Algorithm 1 selects only $\epsilon$-optimal arms of an objective, after a certain rounds $T_\epsilon$.

## 5 Experiment

We conduct experiments in both fixed and stochastic context settings to evaluate the empirical performance of our proposed algorithm MORR-Greedy. We compare the proposed algorithm with the two most well-known multi-objective algorithms P-UCB (Drugan & Nowe, 2013) and MOGLM-UCB (Lu et al., 2019). P-UCB is the first multi-objective algorithm for non-contextual MAB setting, while MOGLM-UCB is developed to solve generalized linear bandit problems. We confirm the performance of the three algorithms in a linear bandit $y_m(t) = \mathcal{N}(x_i^T\theta_m^*, 0.1^2)$ for all $i \in [K]$ and $m \in [M]$. Our results are averaged over 10 different instances for each $(d, K, M)$-combination, and we conducted a 10-round reputation experiment on the same problem instance. Detailed settings of experiments can be found in Appendix G.1.

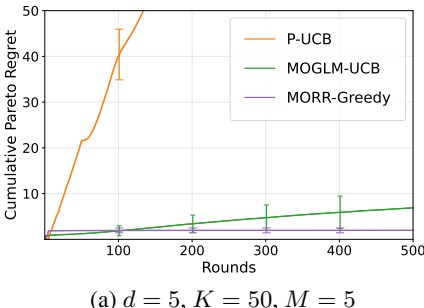
(a) $d = 5$, $K = 50$, $M = 5$

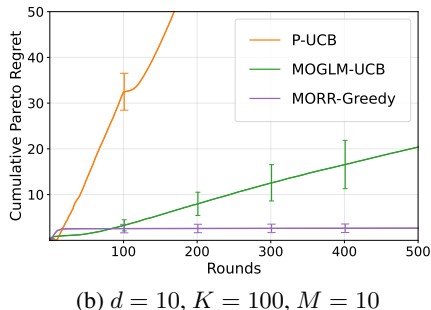
(b) $d = 10$, $K = 100$, $M = 10$

Figure 1: Evaluation of multi-objective bandit algorithms with tuned parameters

Figure 1 illustrates the performance of each algorithm under two different $(d, K, M)$ combinations in the fixed feature setup. Additional results, including performance in various settings and with

stochastic contexts, can be found in Appendix G.1. The results clearly demonstrate that our proposed algorithm outperforms the others empirically, despite its simpler structure. Notably, while the performance of the other algorithms degrades as the number of objectives and arms increases, `MORR-Greedy` maintains consistent performance. This shows that when the objectives are sufficiently diverse, our near-greedy algorithm exhibits superior empirical performance, surpassing more complex methods. Additionally, we conducted experiments to evaluate how our algorithm selects near-optimal arms for each objective in a balanced manner and to assess its performance under various initial objective parameters. The results are presented in Appendices G.2 and G.3.

## 6 CONCLUSION

In this work, we introduced `MORR-Greedy`, a near-greedy algorithm for multi-objective bandits. We identified sufficient conditions where free exploration arises from objective diversity, enabling our algorithm to achieve $\widetilde{\mathcal{O}}(\frac{\sqrt{dT}}{\lambda_0})$ regret bounds under objective diversity and feature regularity. We also introduced the concept of objective fairness, ensuring equal treatment of all objectives, and demonstrated that `MORR-Greedy` satisfies this criterion. Our findings offer a new perspective, showing that diverse objectives can actually enhance learning in multi-objective bandits.

## 7 REPRODUCIBLITY STATEMENT

For theoretical results, we provide all assumptions in Section 4 and a complete proof of our main results in Appendix A. We also present similar results that can be obtained under different environment or assumptions and the proofs of the results in Appendix B, E, and F. We also included the data and code, along with instructions to reproduce our experimental results, in the supplementary material.

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

# Appendix

## Table of Contents

## A  ANALYSIS OF `MORR-Greedy` WITH FIXED FEATURES

### A.1  PROOF OF LEMMA 1

The key idea of our analysis is that, in each cycle in round-robin process, the algorithm selects arms that are close to the directions of $\theta_1^*, \ldots, \theta_M^*$. The following lemma tells us that we can bound the distance between the selected arms and the objective parameters after enough exploration rounds and enables us to derive the diversity of the selected arms from the objective diversity.

**Lemma A.1** (Near Optimal Zone Construction). *Given Assumptions 1, assume the OLS estimator satisfies $||\hat{\theta}_m(s) - \theta_m^*|| \leq \alpha$, for $m \in [M]$ and $s \geq T_0 + 1$. If $x \in \mathbb{B}^d$ satisfies $x^\top \frac{\hat{\theta}_m(s)}{||\hat{\theta}_m(s)||} \leq \gamma$, then the distance between $x$ and $\theta_m^*$ is bounded by*

$$||\theta_m^* - x||_2 \leq \sqrt{2(1 + \alpha\sqrt{1 - \gamma^2} - \gamma\sqrt{1 - \alpha^2})}.$$

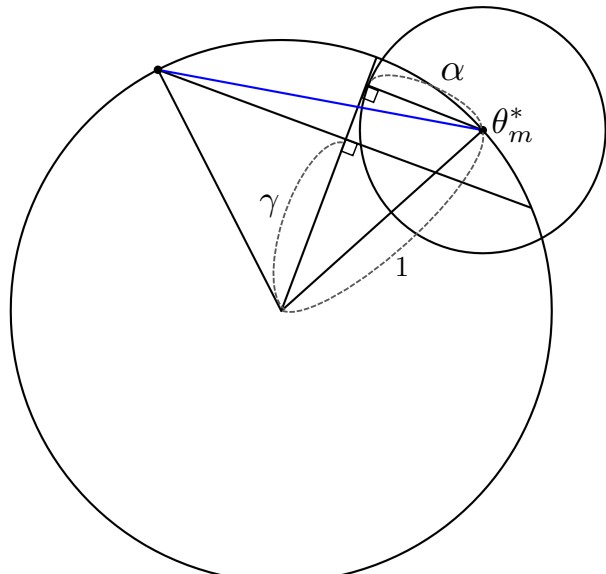

Figure A.1: The larger circle represents the unit sphere in $\mathbb{R}^d$ while the interior of smaller circle indicates the region where $\hat{\theta}_m(s)$ may exist. Then, the blue line illustrates the case when $x$ that satisfies $x^\top \frac{\hat{\theta}_m(s)}{||\hat{\theta}_m(s)||} \geq \gamma$ is farthest from the $\theta_m^*$.

*Proof of Lemma A.1.* Consider the case when $x$ is the farthest from $\theta_m^*$.
 As we easily can see from Figure A.1,

$$||\theta_m^* - x||_2^2 \leq (\alpha + \sqrt{1 - \gamma^2})^2 + (\sqrt{1 - \alpha^2} - \gamma)^2 = 2 + 2\alpha\sqrt{1 - \gamma^2} - 2\gamma\sqrt{1 - \alpha^2}.$$

$\blacksquare$

Now, we will demonstrate that the minimum eigenvalue of the gram matrix increases proportionally with $t$. This technique is often used in the analysis of greedy algorithms for a single objective with stochastic contexts (Kannan et al., 2018; Bastani et al., 2021), where the increase in the minimum eigenvalue of the gram matrix is used to derive the bound of $||\hat{\theta}(t) - \theta_*||_2$. We use the diversity of the objectives to establish a constant lower bound for the minimum eigenvalue of the gram matrix consists of the selected feature vectors within a single cycle of the Round Robin process, i.e. $\lambda_{\min}\left(\sum_{s=t_0}^{t_0+M-1} x(s)x(s)^\top\right)$ for a cycle $s = t_0, t_0 + 1, \ldots, t_0 + M - 1$.

**Lemma A.2.** *Let Assumptions 1, 2, and 3 hold. Assume the OLS estimator satisfies $||\hat{\theta}_m(s) - \theta_m^*|| \leq \alpha_0$, for all $m \in [M]$ and $s \geq T_0 + 1$. Then, the selected arms for a single cycle $s = t_0, t_0 + 1, \ldots, t_0 + M - 1$ ($t_0 > T_0$) by Algorithm1 satisfies*

$$\lambda_{\min}\left(\sum_{s=t_0}^{t_0+M-1} x(s)x(s)^\top\right) \geq \left(\lambda_0 - 2\sqrt{2\left(1 + \alpha_0\sqrt{1 - \gamma_0^2} - \gamma_0\sqrt{1 - \alpha_0^2}\right)}\right) M.$$

*Proof of Lemma A.2.* Let $m(s)$ be the target objective for iteration $s$, and consider near optimal zone $R(s) := \{x \in \mathbb{B}^d \mid x^\top \frac{\hat{\theta}_{m(s)}(s)}{||\hat{\theta}_{m(s)}(s)||} \geq \gamma_0\}$. Since $||\hat{\theta}_{m(s)}(s) - \theta_{m(s)}^*|| \leq \alpha_0$ holds for $s \geq T_0 + 1$, we can easily confirm $\left|\left|\frac{\hat{\theta}_{m(s)}(s)}{||\hat{\theta}_{m(s)}(s)||} - \theta_{m(s)}^*\right|\right| \leq \alpha_0$ holds geometrically. Then, Assumption 3 guarantees that there exists at least 1 arm in $R(s)$, and so $x(s)$ should be in $R(s)$. Thus, by Lemma A.1, we can get $||x(s) - \theta_{m(s)}^*|| \leq \sqrt{2\left(1 + \alpha_0\sqrt{1 - \gamma_0^2} - \gamma_0\sqrt{1 - \alpha_0^2}\right)}$.

Then, for any unit vector $u \in \mathbb{B}^d$,

$$u^\top \left( \sum_{s=t_0}^{t_0+M-1} x(s)x(s)^\top \right) u = \sum_{s=t_0}^{t_0+M-1} \langle u, x(s) \rangle^2$$

$$= \sum_{s=t_0}^{t_0+M-1} \left\langle u, \theta^*_{m(s)} + (x(s) - \theta^*_{m(s)}) \right\rangle^2$$

$$= \sum_{s=t_0}^{t_0+M-1} \left\{ \left\langle u, \theta^*_{m(s)} \right\rangle^2 + \left\langle u, x(s) - \theta^*_{m(s)} \right\rangle^2 + 2 \left\langle u, \theta^*_{m(s)} \right\rangle \left\langle u, x(s) - \theta^*_{m(s)} \right\rangle \right\}$$

$$\geq u^\top \left( \sum_{m=1}^{M} \theta^*_m \left( \theta^*_m \right)^\top \right) u + 0 - 2 \sqrt{2 \left( 1 + \alpha_0 \sqrt{1-\gamma_0^2} - \gamma_0 \sqrt{1-\alpha_0^2} \right)} M$$

$$\geq M\lambda_0 - 2 \sqrt{2 \left( 1 + \alpha_0 \sqrt{1-\gamma_0^2} - \gamma_0 \sqrt{1-\alpha_0^2} \right)} M.$$

Therefore, we have $\lambda_{\min} \left( \sum_{s=t_0}^{t_0+M-1} x(s)x(s)^\top \right) \geq \left( \lambda_0 - 2 \sqrt{2 \left( 1 + \alpha_0 \sqrt{1-\gamma_0^2} - \gamma_0 \sqrt{1-\alpha_0^2} \right)} \right) M.$

■

*Proof of Lemma 1.* If we choose $\lambda$ as stated in Lemma 1, the OLS estimator satisfies $||\hat{\theta}_m(s) - \theta^*_m|| \leq \alpha_0$ for all $s \geq T_0 + 1$ and $m \in [M]$ with probability $1 - 2M\delta$, by Lemma A.4. Thus, by applying Lemma A.2 to every single round after exploration, we have, for $t \geq T_0 + M$,

$$\lambda_{\min} \left( \sum_{s=1}^{t-1} x(s)x(s)^\top \right) \geq \lambda_{\min} \left( \sum_{s=1}^{T_0} x(s)x(s)^\top \right) + \lambda_{\min} \left( \sum_{s=T_0+1}^{t-1} x(s)x(s)^\top \right)$$

$$\geq \lambda + \left[ \frac{t-1-T_0}{M} \right] \times C_0 M,$$

$$\geq \lambda + C_0(t - T_0 - M),$$

where $C_0 := \lambda_0 - 2\sqrt{2(1 + \alpha\sqrt{1-c^2} - c\sqrt{1-\alpha^2})}$.

■

## A.2 Proof of Theorem 1

With Lemma 1, We are ready to derive the $l_2$ bound of $\hat{\theta}_m(t) - \theta^*_m$ for $m \in [M]$.

**Lemma A.3.** *Fix $\delta > 0$. Under the same conditions as Lemma 1, with probability at least $1 - 3M\delta$, for all $m \in [M]$ and $t \geq 2T_0 + 2M$, the OLS estimator $\hat{\theta}_m(t)$ of $\theta^*_m$ satisfies*

$$\left\| \hat{\theta}_m(t) - \theta^*_m \right\|_2 \leq \frac{2\sigma}{C_0} \sqrt{\frac{d \log(dt/\delta)}{t - T_0 - M}},$$

*where $C_0 = \lambda_0 - 2\sqrt{2(1 + \alpha\sqrt{1-c^2} - c\sqrt{1-\alpha^2})}$.*

*Proof of Lemma A.3.* From the closed form of the OLS estimators, for all $m \in [M]$,

$$\left\| \hat{\theta}_m(t) - \theta^*_m \right\|_2 = \left\| \left( \sum_{s=1}^{t-1} x(s)x(s)^\top \right)^{-1} \sum_{s=1}^{t-1} x(s)\eta_{a(s),m}(s) \right\|_2$$

$$\leq \frac{1}{\lambda_{\min} \left( \sum_{s=1}^{t-1} x(s)x(s)^\top \right)} \left\| \sum_{s=1}^{t-1} x(s)\eta_{a(s),m}(s) \right\|_2$$

For the denominator, we have $\lambda_{\min}(V_{t-1}) \geq \lambda + C_0(t - T_0 - M)$ for $t \geq T_0 + M$, with probability at least $1 - 2M\delta$, by Lemma 1. To bound the $l_2$ norm of $S_{t-1,m} := \sum_{s=1}^{t-1} x(s)\eta_{a(s),m}(s)$, we can use Lemma H.1, the martingale inequality of Kannan et al. (2018). The lemma states for fixed $m \in [M]$, $\|S_{t-1,m}\|_2 \leq \sigma\sqrt{2dt\log(dt/\delta)}$ holds with probability at least $1 - \delta$. Therefore, with probability at least $1 - 3M\delta$, for all $m \in [M]$ and $t \geq 2T_0 + 2M$,

$$\left\|\hat{\theta}_m(t) - \theta_m^*\right\|_2 \leq \frac{\sigma\sqrt{2dt\log(dt/\delta)}}{\lambda + C_0(t - T_0 - M)} \leq \frac{2\sigma}{C_0}\sqrt{\frac{d\log(dt/\delta)}{t - T_0 - M}}.$$

The last inequality holds when $t \geq 2T_0 + 2M$.

∎

*Proof of Theorem 1.* Let $E$ be the event that $\left\|\hat{\theta}_m(t) - \theta_m^*\right\|_2 \leq \frac{2\sigma}{C_0}\sqrt{\frac{d\log(dtT)}{t - T_0 - M}}$ holds for all $m \in [M]$ and $t \geq 2T_0 + 2M$ where $C_0 = \lambda_0 - 2\sqrt{2(1 + \alpha\sqrt{1 - c^2} - c\sqrt{1 - \alpha^2})}$. Then, $\mathbb{P}(\bar{E}) \leq \frac{3M}{T}$ by Lemma A.3 with $\delta = \frac{1}{T}$.

Let $m(t)$ be the target objective for round $t$ and $a_m^*$ be the optimal arm with respect to objective $m$. Then, the suboptimality gap on round $t$ is bounded by

$$\Delta_{a(t)}(t) \leq \left(x_{a_{m(t)}^*}\right)^\top \theta_{m(t)}^* - x(t)^\top \theta_{m(t)}^* \leq 2\|\hat{\theta}_{m(t)}(t) - \theta_{m(t)}^*\|_2.$$

Let $\Delta_{\max}$ be the maximum suboptimality gap. For $t \geq 2T_0 + 2M$,

$$\mathbb{E}[\Delta_{a(t)}(t)] \leq \mathbb{E}[\Delta_{a(t)}(t) \mid E] + \mathbb{P}(E)\Delta_{\max}$$

$$\leq 2\mathbb{E}[\|\hat{\theta}_{m(t)}(t) - \theta_{m(t)}^*\|_2 \mid E] + \frac{3M}{T}\Delta_{\max}$$

$$\leq \frac{4\sigma}{C_0}\sqrt{\frac{d\log(dtT)}{t - T_0 - M}} + \frac{3M}{T}\Delta_{\max}.$$

Then, the Pareto Regret is bounded by

$$\mathcal{PR}(T) = \sum_{t=2T_0+2M+1}^{T} \mathbb{E}[\Delta_{a(t)}(t)] + (2T_0 + 2M)\Delta_{\max}$$

$$\leq \sum_{t=2T_0+2M+1}^{T} \frac{4\sigma}{C_0}\sqrt{\frac{d\log(dtT)}{t - T_0 - M}} + \{(\frac{3M}{T})T + 2T_0 + 2M\}\Delta_{\max}$$

$$\leq \frac{4\sigma}{C_0}\sqrt{2d\log(dT)}\int_0^T \frac{1}{\sqrt{t}}dt + \{2T_0 + 5M\}\Delta_{\max}$$

$$\leq \frac{8\sigma}{C_0}\sqrt{2dT\log(dT)} + 2\{2T_0 + 5M\}.$$

The last inequality holds because we have $\Delta_{\max} \leq 2$ under Assumption 1.

∎

### A.3   OBJECTIVE FAIRNESS OF MORR-GREEDY

*Proof of Theorem 2.* Define the event $\Omega_{m,t}$ for all $m \in [M]$ as

$$\Omega_{m,t} := \{\omega \in \Omega \mid \text{Objective } m \text{ is a target objective for round } t\}.$$

Then, $\mathbb{P}(\Omega_{m,t}) = \mathbf{1}_{\{t \equiv m \mod M\}}$ from the Round-Robin process.

Let $E$ be the event that $\left\|\hat{\theta}_m(t) - \theta_m^*\right\|_2 \leq \frac{2\sigma}{C_0}\sqrt{\frac{d\log(dtT)}{t - T_0 - M}}$ holds for all $m \in [M]$ and $t \geq 2T_0 + 2M$ where $C_0 = \lambda_0 - 2\sqrt{2(1 + \alpha\sqrt{1 - c^2} - c\sqrt{1 - \alpha^2})}$. Then, $\mathbb{P}(\bar{E}) \leq \frac{3M}{T}$ by Lemma A.3 with

$\delta = \frac{1}{T}$. We know that on $\Omega_{m,t} \cap E$, for $t \geq 2T_0 + 2m$,

$$\mu_m^* - \mu_{a(t),m} \leq 2||\hat{\theta}_m(t) - \theta_m^*||_2 \leq \frac{4\sigma}{C_0}\sqrt{\frac{d\log(dtT)}{t - T_0 - M}} \leq \frac{4\sigma}{C_0}\sqrt{\frac{2d\log(dT)}{t - T_0 - M}}.$$

Let $T_\epsilon = \max(\lfloor \frac{32\sigma^2 d\log(dT)}{C^2\epsilon^2}\rfloor + T_0 + M, \ 2T_0 + 2M)$. Then, on $\Omega_{m,t} \cap E$, we have $\mu_m^* - \mu_{a(t),m} < \epsilon$ for all $t > T_\epsilon$.

Therefore, for all $m \in [M]$,

$$\frac{1}{T}\mathbb{E}\left[\sum_{t=1}^{T}\mathbf{1}_{\{\mu_m^* - \mu_{a(t),m} < \epsilon\}}\right] = \frac{1}{T}\sum_{t=1}^{T}\mathbb{E}\left[\mathbf{1}_{\{\mu_m^* - \mu_{a(t),m} < \epsilon\}}\right]$$

$$\geq \frac{1}{T}\sum_{t=1}^{T}\mathbb{E}[\mathbf{1}_{\{\mu_m^* - \mu_{a(t),m} < \epsilon\}} \mid \Omega_{m,t}]\,\mathbb{P}(\Omega_{m,t})$$

$$\geq \frac{1}{T}\sum_{t=T_\epsilon+1}^{T}\mathbb{E}[\mathbf{1}_{\{\mu_m^* - \mu_{a(t),m} < \epsilon\}} \mid \Omega_{m,t}]\,\mathbb{P}(\Omega_{m,t})$$

$$\geq \frac{1}{T}\sum_{t=T_\epsilon+1, \ M|t-m}^{T}\mathbb{E}[\mathbf{1}_{\{\mu_m^* - \mu_{a(t),m} < \epsilon\}} \mid \Omega_{m,t} \cap E]\,\mathbb{P}(E)$$

$$\geq \frac{1}{T}\sum_{t=T_\epsilon+1, \ M|t-m}^{T}\mathbb{P}(E)$$

$$\geq \frac{1}{T}\left[\frac{T - T_\epsilon}{M}\right]\left(1 - \frac{3M}{T}\right)$$

$$\geq \left(\frac{T - T_\epsilon - M}{MT}\right)\left(1 - \frac{3M}{T}\right)$$

$\blacksquare$

### A.4 THE PARAMETER $\lambda$ AND THE NUMBER OF INITIAL ROUNDS

Now, let's discuss the appropriate value of $\lambda$, the threshold of the minimum eigenvalue of the gram matrix. For convenience, denote $V_t := \sum_{s=1}^{t} x(s)x(s)^\top$ and $S_t := \sum_{s=1}^{t} x(s)\eta_{a(s)}(s)^\top$. When the minimum eigenvalue of the empirical covariance matrix $V_{T_0}$ exceeds a certain threshold, we can guarantee the $l_2$ bound of the OLS estimator $\hat{\theta}(t)$ of $\theta_*$ for $t > T_0$ with high probability. I.e.,

$$\lambda_{\min}(V_{T_0}) \geq f(a) \ \Rightarrow \ \left|\left|\hat{\theta}(t) - \theta_*\right|\right|_2 \leq a, \ \ \forall t \geq T_0$$

If we set $\lambda = f(\alpha_0)$, then with high probability, $\left|\left|\hat{\theta}_m(t) - \theta_m^*\right|\right| \leq \alpha_0$ after playing with initial values $\beta_0, \ldots, \beta_M$.

Kveton et al. (2020) suggest $f(a)$ using a bound of $||S_t||_{V_{t-1}^{-1}}$. However, a small mistake was made in their process: the bound they derived by modifying Theorem 1 of Abbasi-Yadkori et al. (2011) is actually a bound for $||\sum_{s=\tau_0+1}^{t} x(s)\eta_{a(s)}(s)^\top||_{V_{t-1}^{-1}}$, where $\tau_0 = \min\{t \geq 1 : V_t \succ 0\}$, not $||S_t||_{V_{t-1}^{-1}}$. To address this problem, the simplest approach would be to use the bound of $||S_t||_2$ suggested by Kannan et al. (2018). In this case, $f(a)$ would include a term with $\sqrt{dT}$. Alternatively, using the bound of $||S_t||_{V_{t-1}^{-1}}$ proposed by Li et al. (2017), that $f(a)$ can be constructed without $\sqrt{T}$, relying on $d\log T$. Through these two approaches, we can obtain an $\widetilde{\mathcal{O}}(\min(\sqrt{dT}, d\log T))$ bound for $\lambda$.

**Lemma A.4.** *Given Assumption 1, for any $a > 0$ and $\delta > 0$, if we set $\lambda$*

$$\lambda = \min\left[\frac{\sigma}{a}\sqrt{2dT\log(\frac{dT}{\delta})}, \ \frac{4\sigma^2}{a^2}\left(\frac{d}{2}\log\left(1 + \frac{2T}{d}\right) + \log\left(\frac{1}{\delta}\right)\right)\right],$$

*then with probability at least $1 - 2M\delta$, the OLS estimator satisfies $||\hat{\theta}_m(t) - \theta_m^*||_2 \le a$ for all $m \in [M]$ and $t \ge T_0 + 1$.*

*Proof of Lemma A.4.* First we will bound $\lambda$ using the fact

$$\left|\left|\hat{\theta}_m(t) - \theta_m^*\right|\right|_2 = \left|\left|(V_{t-1})^{-1}S_{t-1,m}\right|\right| \le \frac{1}{\lambda_{\min}(V_{t-1})}||S_{t-1,m}||_2,$$

where $S_{t,m} := \sum_{s=1}^{t} x(s)\eta_{a(s),m}(s)^\top$.

Since $||S_{t-1,m}||_2 \le \sigma\sqrt{2dt\ln(td/\delta)}$ holds with probability at least $1 - \delta$ by Lemma H.1 and $\lambda_{\min}(V_{t-1}) \ge \lambda_{\min}(V_{T_0})$ for $t \ge T_0 + 1$, we have $\left|\left|\hat{\theta}_m(t) - \theta_m^*\right|\right|_2 \le a$ for all $m \in [M]$ and $t \ge T_0 + 1$ with probability at least $1 - M\delta$ when the value of $\lambda$ set to $\frac{\sigma}{a}\sqrt{2dT\log(\frac{dT}{\delta})}$.

Alternatively, we can use the fact

$$\left|\left|\hat{\theta}_m(t) - \theta_m^*\right|\right|_2^2 = (S_{t-1,m})^\top V_{t-1}^{-1}V_{t-1}^{-1}S_{t-1,m} \le \frac{1}{\lambda_{\min}(V_{t-1})}||S_{t-1,m}||_{V_{t-1}^{-1}}^2.$$

By Lemma H.2, $||S_{t-1,m}||_{V_{t-1}^{-1}}^2 \le 4\sigma^2(\frac{d}{2}\log(1 + \frac{2t}{d}) + \log(\frac{1}{\delta}))$ holds with probability at least $1 - \delta$, and hence, we have $\left|\left|\hat{\theta}_m(t) - \theta_m^*\right|\right|_2 < a$ for all $m \in [M]$ and $t \ge T_0 + 1$ with probability at least $1 - M\delta$ by setting $\lambda$ to $\frac{4\sigma^2}{a^2}(\frac{d}{2}\log(1 + \frac{2T}{d}) + \log(\frac{1}{\delta}))$.

∎

*Proof of Corollary 1.* From the definition of the exploration facilitating initial objective parameters, the minimum eigenvalue of the gram matrix consists of the feature vectors greedily selected by exploration facilitating initial objective parameters satisfies

$$\lambda_{\min}\left(\sum_{s=t_0}^{t_0+M-1} x(s)x(s)^\top\right) \ge \left(\lambda_0 - 2\sqrt{2\left(1 + \alpha_0\sqrt{1 - \gamma_0^2} - \gamma_0\sqrt{1 - \alpha_0^2}\right)}\right)M,$$

by Lemma A.2. Then, for any $T_1 \ge \lfloor\frac{\lambda}{C_1}\rfloor + M$, if we keep playing with the initial values for $T_1$ rounds,

$$\lambda_{\min}\left(\sum_{s=1}^{T_1} x(s)x(s)^\top\right) \ge \left\lceil\frac{T_1}{M}\right\rceil \times C_1 M \ge C_1(T_1 - M) \ge \lambda.$$

Hence, we have $T_0 \le \lfloor\frac{\lambda}{C_1}\rfloor + M$.

∎

# B   ANALYSIS OF MORR-GREEDY WITH STOCHASTIC CONTEXTS

## B.1   SETTING

In each round $t = 1, 2, \ldots, T$, the set of feature vectors $\chi(t) = \{x_i(t) \in \mathbb{R}^d, i \in [K]\}$ is drawn from some unknown distribution $P_\chi(t)$. Each arm's feature $x_i(t) \in \chi(t)$ for $i \in [K]$ need not be independent of each other and can possibly be correlated. In this case, we denote $x_{a(t)}(t)$ as $x(t)$. Other settings are identical to the fixed arms case.

## B.2   RESULTS FOR STOCHASTIC CONTEXTS

In this section, we first present the regret bound of MORR-Greedy and the results on objective fairness when played in a stochastic context. The proofs of each theorem are provided in the subsequent sections.

To analyze the `MORR-Greedy` algorithm in the case of stochastic contexts, it is necessary to modify the definition of $\gamma$-regularity slightly. In the stochastic version, it is necessary to assume the existence of a near-optimal arm not only near the true objective parameter but also in all directions to ensure that the selected arms during the exploration process are sufficiently diverse. Instead, there is a clear advantage in that, even with the assumption of a small positive bound on the probability of the existence of near-optimal arms allows us to obtain a regret bound that differs only by a constant factor in the results under fixed contexts.

**Definition 7** ($\gamma$-Regular condition for Stochastic Contexts). *For fixed $\gamma \leq 1$, we say that the distribution $P_\chi(t)$ of feature vector set $\chi(t)$ satisfies $\gamma$-Regular condition if there exists a positive number $q_\gamma$ that satisfies*

$$\forall \beta \in \mathbb{S}^{d-1}, \; \mathbb{P}_{\chi(t)}[\exists i \in [K], \; x_i(t)^\top \beta \geq \gamma] \geq q_\gamma.$$

**Assumption B.1** ($\gamma$-Regularity for Stochastic Contexts). *We assume $P_\chi(t)$ satisfies $\gamma_0$-regular condition for all $t \in [T]$, with $\gamma_0 > 1 - \frac{\lambda_0^2}{8}$ where $\lambda_0 = \lambda_{\min}(\frac{1}{M} \sum_{m=1}^M \theta_m^* (\theta_m^*)^\top)$.*

In the following analysis, we use notation $q_0 := q_{\gamma_0}$ where $q_\gamma$ satisfies the condition in Definition 7. The fixed version of $\gamma$-regularity can be viewed as a special case of Definition 7 with $q_\gamma = 1$. As A detailed analysis of $\gamma$-regularity is provided in Appendix C. As stated earlier in Section 4, $\gamma$-regularity is a condition that applies the notion of $\beta$-regularity from Bayati et al. (2020) to context distribution in the multi-objective setting. We explain why $\gamma$-regularity can be treated as a weaker condition than $\beta$-regularity in Appendix C.2

The following assumption is essential to guarantee that in each round $t$, the feature vectors drawn from $P_\chi(t)$ are not influenced by previous rounds $s = 1, \ldots, t-1$.

**Assumption B.2** (Independently Distributed Contexts). *The context sets $\chi(1), \ldots, \chi(T)$, drawn from unknown distribution $P_\chi(1), \ldots, P_\chi(T)$ respectively, are independently distributed across time.*

All of the greedy linear contextual bandit with stochastic contexts assumes the independence of context sets. It is important to note that feature vectors within the same round are allowed to be dependent, even under Assumption B.2. Additionally, this independence assumption does not imply that the feature vector is diverse. For example, assuming independence does not ensure the diversity of the feature vector if the distribution of feature vectors only contains two candidates.

The following theorem demonstrates that the `MORR-Greedy` algorithm also possesses optimal regret in the case of stochastic contexts by replacing Assumption 3 with Assumptions 4 and 5. The leading term is $2/q_0$ times of that of the result from fixed contexts.

**Theorem B.1** (Pareto Regret of `MORR-Greedy` with Stochastic Contexts). *Suppose Assumptions 1, 2, B.1, and B.2 hold. If we run Algorithm 1 with $\lambda = \min\left[\frac{\sigma}{\alpha_0}\sqrt{2dT\log(dT^2)}, \; \frac{4\sigma^2}{\alpha_0^2}\left(\frac{d}{2}\log\left(1 + \frac{2T}{d}\right) + \log(T)\right)\right]$ for some $\alpha_0 < \sqrt{\frac{\lambda_0^2}{4} - \frac{\lambda_0^4}{64}} \; \gamma_0 - \left(1 - \frac{\lambda_0^2}{8}\right)\sqrt{1 - \gamma_0^2}$, the Pareto regret of Algorithm 1 is bounded by*

$$\mathcal{PR}(T) \leq C_2\sqrt{2dT\log(dT)} + 2\left(2T_0 + 5M + \frac{10d}{Cq_0}\right),$$

*where $C_2 = \frac{16\sigma}{\lambda_0 q_0 - 2q_0\sqrt{2\left(1 + \alpha_0\sqrt{1 - \gamma_0^2} - \gamma_0\sqrt{1 - \alpha_0^2}\right)}}$.*

The proof of the theorem is given in Appendix B.3.
We confirmed that the `MORR-Greedy` algorithm also satisfies the objective fairness in the case of stochastic contexts.

**Theorem B.2** (Objective Fairness of `MORR-Greedy` with Stochastic Contexts). *Given Assumptions 1, 2, B.1, and B.2, the Algorithm 1 satisfies for all $m \in [M]$,*

$$\frac{1}{T}\mathbb{E}\left[\sum_{t=1}^T \mathbf{1}_{\{\mu_m^* - \mu_{a(t),m} < \epsilon\}}\right] \geq \left(\frac{T - T_\epsilon - M}{MT}\right)\left(1 - \frac{3M}{T} - d\left(\frac{1}{dT}\right)^{\frac{64\sigma^2 d}{Cq_0\epsilon^2}}\right),$$

*where $T_\epsilon = \max(\lfloor \frac{64\sigma^2 d\log(dT)}{(\lambda_0 - 2\sqrt{2\left(1 + \alpha_0\sqrt{1 - \gamma_0^2} - \gamma_0\sqrt{1 - \alpha_0^2}\right)})^2 q_0^2 \epsilon^2}\rfloor + T_0 + M, \; 2T_0 + 2M)$ in the same setting as Theorem B.1.*

The proof of the theorem is given in Appendix B.4.

**Remark B.1.** *Note that the $p_{\epsilon,T} \geq \left(\frac{T-T_\epsilon-M}{MT}\right)\left(1 - \frac{3M}{T} - d(\frac{1}{dT})^{\frac{64\sigma^2 d}{Cq_0\epsilon^2}}\right)$ and the lower bound*

*satisfies $\left(\frac{T-T_\epsilon-M}{MT}\right)\left(1 - \frac{3M}{T} - d(\frac{1}{dT})^{\frac{64\sigma^2 d}{Cq_0\epsilon^2}}\right) \to \frac{1}{M}$ as $T \to \infty$.*

### B.3 Proofs of Theorem B.1

In the stochastic version, similarly to the fixed version, we can establish a constant lower bound for the minimum eigenvalue of the Gram matrix formed by the selected feature vectors within a single cycle of the Round Robin process. In the case of fixed contexts, Lemma A.2 demonstrates that the eigenvalue of the Gram matrix can increase by a constant amount (or more) during a single round. The following lemma is a modified version of Lemma A.2, adapted to apply to the situation of stochastic contexts.

**Lemma B.1** (Near Optimal Zone Construction). *Given Assumptions 1 and 2, assume the OLS estimator satisfies $||\hat{\theta}_m(s) - \theta_*^j|| \leq \alpha$, for all $m \in [M]$ and $s \geq T_0 + 1$. Define near optimal zone $R_m(s)$ with respect to obejective $j$ on round $s$ as*

$$R_m(s) = \{x \in \mathbb{B}^d \mid x^\top \frac{\hat{\theta}_m(s)}{||\hat{\theta}_m(s)||} \geq \gamma\}.$$

*Let $z_1, \ldots, z_M$ be any vectors in near optimal zones $R_1(t_1), \ldots, R_M(t_M)$ of different objectives on round $t_1, \ldots, t_m$ where $t_j \geq T_0 + 1$ for all $m \in [M]$. Then, the minimum eigenvalue of Gram matrix consists of $z_1, \ldots, z_M$ satisfies*

$$\lambda_{\min}\left(\sum_{m=1}^M z_m(z_m)^\top\right) \geq \left(\lambda_0 - 2\sqrt{2\left(1 + \alpha\sqrt{1-\gamma^2} - \gamma\sqrt{1-\alpha^2}\right)}\right).$$

The proof can be demonstrated in the same manner as Lemma A.2.

**Remark B.2.** *The Lemma suggests when the algorithm can quit the exploration to obtain the linear increase of the minimum eigenvalue of Gram matrix. If the algorithm explores until $\hat{\theta}_m(t)$ is within $\gamma$ of $\theta_m^*$ for $\alpha < \sqrt{\frac{\lambda_0^2}{4} - \frac{\lambda_0^4}{64}}\gamma - \left(1 - \frac{\lambda_0^2}{8}\right)\sqrt{1-\gamma^2}$, then we have a positive value for $\lambda_0 - 2\sqrt{2\left(1 + \alpha\sqrt{1-\gamma^2} - \gamma\sqrt{1-\alpha^2}\right)}$.*

The next step is proving the constant increase of the minimum eigenvalue of Gram matrix in a single round. We will make a constant lower bound for $\lambda_{\min}(\sum_{s=t_0}^{t_0+M-1}\mathbb{E}[x(s)x(s)^\top|\mathcal{H}_{s-1}])$ within a single cycle of the Round Robin process.

**Lemma B.2.** *Suppose Assumptions 1, 2, B.1, and B.2 hold. Assume the OLS estimator satisfies $||\hat{\theta}_m(s)) - \theta_m^*|| \leq \alpha_0$, for all $m \in [M]$ and $s \geq T_0 + 1$ for some $\alpha_0 > 0$. Then, the selected arms for a single cycle $s = t_0, t_0 + 1, \ldots, t_0 + M - 1$ $(t_0 > T_0)$ by Algorithm1 satisfies*

$$\lambda_{\min}(\sum_{s=t_0}^{t_0+M-1}\mathbb{E}[x(s)x(s)^\top|\mathcal{H}_{s-1}]) \geq \left(\lambda_0 - 2\sqrt{2\left(1 + \alpha_0\sqrt{1-\gamma_0^2} - \gamma_0\sqrt{1-\alpha_0^2}\right)}\right)q_0 M.$$

*Proof of Lemma B.2.* For $s \geq T_0 + 1$, let $m(s)$ be the target objective for iteration $s$ and $R(s) := \{x \in \mathbb{B}^d \mid x^\top \frac{\hat{\theta}_{m(s)}(s)}{||\hat{\theta}_{m(s)}(s)||} \geq \gamma_0\}$. Then,

$$\mathbb{E}[x(s)x(s)^\top|\mathcal{H}_{s-1}]$$
$$\succeq \mathbb{E}[x(s)x(s)^\top|\mathcal{H}_{s-1}, x(s) \in R(s)]\,\mathbb{P}[x(s) \in R(s)|\mathcal{H}_{s-1}]$$
$$= \mathbb{E}[x(s)x(s)^\top|\mathcal{H}_{s-1}, x(s) \in R(s)]\,\mathbb{P}[\exists i \in [K], \, x_i(s) \in R(s)|\mathcal{H}_{s-1}]$$
$$= \mathbb{E}[x(s)x(s)^\top|\mathcal{H}_{s-1}, x(s) \in R(s)]\,\mathbb{P}[\exists i \in [K], \, x_i(s) \in R^{(}s)] \qquad (\because \text{Assumption B.2})$$
$$\succeq q_0\,\mathbb{E}[x(s)x(s)^\top|\mathcal{H}_{s-1}, x(s) \in R(s)] \qquad\qquad (\because \text{Assumption B.1})$$
$$\succeq q_0\,\mathbf{x}(s)\mathbf{x}(s)^\top, \text{ where } \mathbf{x}(s) = \mathbb{E}[x(s)|\mathcal{H}_{s-1}, x(s) \in R(s)]. \qquad (\because \text{Lemma H.4}).$$

Thus, we have $\sum_{s=t_0}^{t_0+M-1} \mathbb{E}[x(s)x(s)^\top | \mathcal{H}_{s-1}] \succeq q_0 \sum_{s=t_0}^{t_0+M-1} \mathbf{x}(s)\mathbf{x}(s)^\top$, where $\mathbf{x}(s) = \mathbb{E}[x(s)|\mathcal{H}_{s-1}, x(s) \in R(s)]$. Since $R(s)$ is a convex set for all $m \in [M]$ and $s > T_0$, $\mathbf{x}(s)$ should be inside $R(s)$, so we can apply Lemma B.1 by

$$\lambda_{\min}(\sum_{s=t_0}^{t_0+M-1} \mathbb{E}[x(s)x(s)^\top | \mathcal{H}_{s-1}]) \geq q_0 \, \lambda_{\min}(\sum_{s=t_0}^{t_0+M-1} \mathbf{x}(s)\mathbf{x}(s)^\top)$$

$$\geq q_0 \times \left( \lambda_0 - 2\sqrt{2\left(1 + \alpha_0\sqrt{1-\gamma_0^2} - \gamma_0\sqrt{1-\alpha_0^2}\right)} \right) M.$$

∎

The following lemma shows that the minimum eigenvalue of Gram matrix increases linearly with $t$ with high probability.

**Lemma B.3** (Minimum eigenvalue growth of Gram matrix with stochastic contexts). *Suppose Assumptions 1, 2, B.1, and B.2 hold. Assume the OLS estimator satisfies $\|\hat{\theta}_m(s) - \theta_m^*\| \leq \alpha_0$ for all $m \in [M]$ and $s \geq T_0 + 1$ for some $\alpha_0 > 0$. Then for $t \geq T_0 + M$, the following holds for the minimum eigenvalue of the Gram matrix of arms selected by Algorithm 1*

$$\mathbb{P}\left[ \lambda_{\min}(\sum_{s=1}^{t-1} x(s)x(s)^\top) \leq \lambda + \frac{C_0 q_0}{2}(t - T_0 - M) \right] \leq d e^{\frac{-C_0 q_0 (t - T_0 - M)}{10}},$$

*where $C_0 = \left( \lambda_0 - 2\sqrt{2\left(1 + \alpha_0\sqrt{1-\gamma_0^2} - \gamma_0\sqrt{1-\alpha_0^2}\right)} \right)$.*

*Proof of Lemma B.3.* By Lemma B.2, for $t \geq T_0 + M$,

$$\lambda_{\min}(\sum_{s=T_0+1}^{t-1} \mathbb{E}[x(s)x(s)^\top | \mathcal{H}_{s-1}]) \geq [\frac{t-1-T_0}{M}] \times C_0 q_0 M \geq (t - T_0 - M)C_0 q_0.$$

In other words, $\mathbb{P}[\lambda_{\min}(\sum_{s=1}^{t-1} \mathbb{E}[x(s)x(s)^\top | \mathcal{H}_{s-1}]) \geq C_0 q_0 (t - T_0 - M)] = 1$ holds for $t \geq T_0 + M$. By applying LemmaH.3 to compute the lower bound of the minimum eigenvalue of the Gram matrix after exploration, we have

$$\mathbb{P}\left[ \lambda_{\min}(\sum_{s=T_0+1}^{t-1} x(s)x(s)^\top) \leq \frac{C_0 q_0}{2}(t - T_0 - M) \right] \leq d\left(\frac{e^{0.5}}{0.5^{0.5}}\right)^{C_0 q_0 (t-T_0-M)} \leq d e^{\frac{-C_0 q_0 (t - T_0 - M)}{10}}.$$

Therefore, by subadditivity of minimum eigenvalue,

$$\mathbb{P}\left[ \lambda_{\min}(\sum_{s=1}^{t-1} x(s)x(s)^\top) \leq \lambda + \frac{C_0 q_0}{2}(t - T_0 - M) \right] \leq d e^{\frac{-C_0 q_0 (t - T_0 - M)}{10}}.$$

The next lemma provides $l_2$ bound of $\hat{\theta}_m(t) - \theta_m^*$ for $m \in [M]$.

**Lemma B.4.** *Suppose Assumptions 1, 2, B.1, and B.2 hold, and fix $\delta > 0$. If we run Algorithm 1 with $\lambda = \min\left[ \frac{\sigma}{\alpha_0}\sqrt{2dT\log(\frac{dT}{\delta})}, \ \frac{4\sigma^2}{\alpha_0^2}\left(\frac{d}{2}\log\left(1 + \frac{2T}{d}\right) + \log\left(\frac{1}{\delta}\right)\right) \right]$ for some $\alpha_0 < \sqrt{\frac{\lambda_0^2}{4} - \frac{\lambda_0^4}{64}} \, \gamma_0 - \left(1 - \frac{\lambda_0^2}{8}\right)\sqrt{1-\gamma_0^2}$, then with probability at least $1 - 3M\delta - d e^{\frac{-C_0 q_0 (t - T_0 - M)}{10}}$, for all $m \in [M]$ and $t \geq 2T_0 + 2M$, the OLS estimator $\hat{\theta}_m(t)$ of $\theta_m^*$ satisfies*

$$\left\| \hat{\theta}_m(t) - \theta_m^* \right\|_2 \leq \frac{4\sigma}{C_0 q_0}\sqrt{\frac{d\log(dt/\delta)}{t - T_0 - M}},$$

*where $C_0 = \lambda_0 - 2\sqrt{2\left(1 + \alpha_0\sqrt{1-\gamma_0^2} - \gamma_0\sqrt{1-\alpha_0^2}\right)}$.*

The proof can be demonstrated in the same manner as Lemma A.3. *Proof of Theorem B.1.* Let $E$ be the event that $||\hat{\theta}^m(t) - \theta_*^j|| \leq \frac{4\sigma}{C_0 q_0}\sqrt{\frac{d\log(dtT)}{t - T_0 - M}}$ holds for all $t > 2T_0 + 2M$ and $m \in [M]$. Then, $\mathbb{P}(\bar{E}) \leq \frac{3m}{T} + de^{\frac{-C_0 q_0(t - T_0 - M)}{10}}$ by Lemma B.4.

Let $\Delta_{\max}$ be the maximum suboptimality gap. For $t \geq 2T_0 + 2m$,

$$\mathbb{E}[\Delta_{a(t)}(t)] \leq \mathbb{E}[\Delta_{a(t)}(t) \mid E] + \mathbb{P}(E)\Delta_{\max}$$

$$\leq 2\mathbb{E}[\,||\hat{\theta}_{m(t)}(t) - \theta_{m(t)}^*||_2 \mid E] + \left(\frac{3M}{T} + de^{\frac{-C_0 q_0(t - T_0 - M)}{10}}\right)\Delta_{\max}$$

$$\leq \frac{8\sigma}{C_0 q_0}\sqrt{\frac{d\log(dtT)}{t - T_0 - M}} + \left(\frac{3m}{T} + de^{\frac{-C_0 q_0(t - T_0 - M)}{10}}\right)\Delta_{\max}.$$

Then, the Pareto Regret is bounded by

$$\mathcal{PR}(T) = \sum_{t=2T_0+2M+1}^{T} \mathbb{E}[\Delta_{a(t)}(t)] + (2T_0 + 2M)\Delta_{\max}$$

$$\leq \frac{8\sigma}{C_0 q_0}\sqrt{\frac{d\log(dtT)}{t - T_0 - M}} + \{(\frac{3M}{T})T + \sum_{t=2T_0+2M+1}^{T} de^{\frac{-C_0 q_0(t - T_0 - M)}{10}} + 2T_0 + 2M\}\Delta_{\max}$$

$$\leq \frac{8\sigma}{C_0 q_0}\sqrt{2d\log(dT)}\int_0^T \frac{1}{\sqrt{t}}dt + \left(2T_0 + 5M + \sum_{t=2T_0+2M}^{T} de^{\frac{-C_0 q_0(t - T_0 - M)}{10}}\right)\Delta_{\max}$$

$$\leq \frac{16\sigma}{C_0 q_0}\sqrt{2dT\log(dT)} + \left(2T_0 + 5M + \frac{10d}{C_0 q_0}\right)\Delta_{\max}$$

$$\leq \frac{16\sigma}{C_0 q_0}\sqrt{2dT\log(dT)} + 2\left(2T_0 + 5M + \frac{10d}{C_0 q_0}\right).$$

The last inequality holds because we have $\Delta_{\max} \leq 2$ under Assumption 1.

∎

### B.4 Proofs of Theorem B.2

*Proof of Theorem B.2.* Define the event $\Omega_{m,t}$ for all $m \in [M]$ as

$$\Omega_{m,t} := \{\omega \in \Omega \mid \text{Objective } m \text{ is a target objective for round } t\}.$$

Then, $\mathbb{P}(\Omega_{m,t}) = \mathbf{1}_{\{t \equiv m \mod M\}}$ from the Round-Robin process.

Let $E$ be the event that $\left\|\hat{\theta}_m(t) - \theta_m^*\right\|_2 \leq \frac{4\sigma}{C_0 q_0}\sqrt{\frac{d\log(dtT)}{t - T_0 - M}}$ holds for all $m \in [M]$ and $t \geq 2T_0 + 2M$ where $C_0 = \lambda_0 - 2\sqrt{2\left(1 + \alpha_0\sqrt{1 - \gamma_0^2} - \gamma_0\sqrt{1 - \alpha_0^2}\right)}$. Then, $\mathbb{P}(\bar{E}) \leq \frac{3M}{T} + de^{-C_0 q_0(t - T_0 - M)}$ by Lemma B.4 with $\delta = \frac{1}{T}$. We know that on $\Omega_{m,t} \cap E$, for $t \geq 2T_0 + 2M$,

$$\mu_m^* - \mu_{a(t),m} \leq 2||\hat{\theta}_m(t) - \theta_m^*||_2 \leq \frac{8\sigma}{C_0 q_0}\sqrt{\frac{d\log(dtT)}{t - T_0 - M}} \leq \frac{8\sigma}{C_0 q_0}\sqrt{\frac{2d\log(dT)}{t - T_0 - M}}.$$

Let $T_\epsilon = \max(\lfloor\frac{64\sigma^2 d\log(dT)}{C_0^2 q_0^2 \epsilon^2}\rfloor + T_0 + M, \ 2T_0 + 2M)$. Then, on $\Omega_{m,t} \cap E$, we have $\mu_m^* - \mu_{a(t),m} < \epsilon$ for all $t > T_\epsilon$.

Therefore, for all $m \in [M]$,

$$\frac{1}{T}\mathbb{E}\left[\sum_{t=1}^{T}\mathbf{1}_{\{\mu_m^* - \mu_{a(t),m} < \epsilon\}}\right] \geq \frac{1}{T}\sum_{t=1}^{T}\mathbb{E}[\mathbf{1}_{\{\mu_m^* - \mu_{a(t),m} < \epsilon\}} \mid \Omega_{m,t}]\,\mathbb{P}(\Omega_{m,t})$$

$$\geq \frac{1}{T}\sum_{t=T_\epsilon+1,\,M|t-m}^{T}\mathbb{E}[\mathbf{1}_{\{\mu_m^* - \mu_{a(t),m} < \epsilon\}} \mid \Omega_{m,t} \cap E]\,\mathbb{P}(E)$$

$$\geq \frac{1}{T}\sum_{t=T_\epsilon+1,\,M|t-m}^{T}\mathbb{P}(E)$$

$$\geq \frac{1}{T}\sum_{t=T_\epsilon+1,\,M|t-m}^{T}\left(1 - \frac{3M}{T} - de^{-C_0 q_0 (T_\epsilon - T_0 - M)}\right)$$

$$\geq \frac{1}{T}\left[\frac{T-T_\epsilon}{M}\right]\left(1 - \frac{3M}{T} - d(\frac{1}{dT})^{\frac{64\sigma^2 d}{C_0 q_0 \epsilon^2}}\right)$$

$$\geq \left(\frac{T-T_\epsilon - M}{MT}\right)\left(1 - \frac{3M}{T} - d(\frac{1}{dT})^{\frac{64\sigma^2 d}{C_0 q_0 \epsilon^2}}\right).$$

■

## C  $\gamma$-REGULARITY

In this section, we present the meaning of $\gamma$-regularity, compare it with the different regularity condition used in another greedy bandit study(Bayati et al. (2020)), and explain the difference between $\gamma$-regularity and context diversity. As mentioned in the previous section, the fixed version can be viewed as a case where the probability $q_\gamma$ is set to 1 in the stochastic version. Therefore, we conducted a general analysis of the $\gamma$ regularity in the stochastic version. (See Definition 7).

### C.1  INTERPRETATION OF $\gamma$-REGULARITY

In summary, $\gamma$-regularity signifies that for any direction $\beta \in \mathbb{S}^{d-1}$, there exists at least one near optimal arm satisfying $x_i(t)^\top \beta \geq \gamma$ with a probability of at least $q_\gamma$. Intuitively, if the union of the supports of each arm $x_i(t)$ for $i \in [K]$ cover all $\mathbb{S}^{d-1}$, $\gamma$-regularity will be guaranteed for all $\gamma < 1$. The following lemma formalizes this concept.

**Lemma C.1.** *Suppose $\chi(t)$ contains $K$ continuous variables $x_1(t), \ldots, x_K(t)$ with density function $f_1, \ldots, f_K$. If $f = f_1 + \ldots + f_K$ is a bounded function and positive near $\mathbb{S}^{d-1}$ (i.e., there exist a radius $r \in (0,1)$ satisfies $f$ is always positive at $\{x \in \mathbb{R}^d \mid r < ||x||_2 < 1\}$), then $P_{\chi(t)}$ satisfies $\gamma$-regularity for all $\gamma \in (0,1)$.*

*Proof of Lemma C.1.* Fix $\gamma \in (0,1)$. From the definition of $f$, $f/K$ is the probability density function of $X = uniform(x_1(t), \ldots, x_K(t))$. Define $p_\beta = \mathbb{P}_{\chi(t)}[X^\top \beta \geq \gamma]$ for unit vector $\beta \in \mathbb{S}^{d-1}$. Then,

$$p_\beta = \mathbb{P}_{\chi(t)}[X^\top \beta \geq \gamma] = \int_{\{x \in \mathbb{B}^R \mid x^\top \beta \geq \gamma\}} \frac{f(x)}{K}dx \geq \int_{\{x \in \mathbb{B}^R \mid x^\top \beta \geq \max(\gamma, r)\}} \frac{f(x)}{K}dx > 0,$$

for all $\beta \in \mathbb{S}^{d-1}$.

Consider the function $F : \beta \xrightarrow{F} p_\beta$. From the boundedness of $f$, we can easily check $F$ is continuous. By the fact that the compactness is preserved by continuous functions, $\{p_\beta \mid \beta \in \mathbb{S}^{d-1}\}$ is compact. Define $q_\gamma := \min\{p_\beta | \beta \in \mathbb{S}^{d-1}\}$, then we have $q_\gamma > 0$ since $p_\beta > 0$ for all $\beta \in \mathbb{S}^{d-1}$. Then, for all $\beta \in \mathbb{S}^{d-1}$

$$\mathbb{P}_{\chi(t)}[\exists i \in [K],\ x_i(t)^\top \beta \geq \gamma] \geq \mathbb{P}_{\chi(t)}[X^\top \beta \geq \gamma] = p_\beta \geq q_\gamma$$

■

**Remark C.1.** *The above lemma states that if the arm set includes just a single continuous variable that can cover $\mathbb{S}^{d-1}$, then $\gamma$-regularity will hold for all $\gamma < 1$ regardless of the distributions of the remaining arms.*

## C.2  $\gamma$-REGULARITY VS $\beta$-REGULARITY

In Bayati et al. (2020), they assume the prior distribution $\Gamma$ of the expected reward $\mu$ of each arm satisfies $\mathbb{P}_\mu[\mu > 1 - \epsilon] = \Theta(\epsilon^\beta)$ for all $\epsilon > 0$ in non-contextual MAB setting. Let's compare this with $\gamma$-regularity when $m = d = 1$. We claim that $\gamma$-regularity can be considered weaker than $\beta$-regularity from three perspectives.

The most significant difference is that in $\beta$-regularity, the probability that the expected reward $\mu_i$ exceeds $1 - \epsilon$ is required for all arm $i \in [K]$, along with the assumption that $\mu_i$'s are drawn independently from prior $\Gamma$. In contrast, in $\gamma$-regularity, it is sufficient to ensure that the probability that one of the $K$ arms satisfies $x_i(t)^\top \beta \geq \gamma$, without the need for the independence assumption between arm vectors. Secondly, unlike $\beta$-regularity, $\gamma$-regularity does not require a specific relationship like $\Theta(1 - \gamma)$ between the probability of the existence of near-optimal arms $\mathbb{P}_{\chi(t)}[\exists i \in [K], \ x_i(t)^\top \beta \geq \gamma]$ and the threshold $\gamma$ ; instead, it focuses on the existence of a positive lower bound $q_\gamma$. Lastly, in Bayati et al. (2020), the $\beta$-regularity assumes the probability of $\mu > 1 - \epsilon$ for all $\epsilon > 0$. In contrast, this study does not mandate $\gamma$-regularity for $\gamma$ very close to 1; it is sufficient to hold $\gamma$-regularity only for $\gamma \geq 1 - (\frac{\lambda_0}{8})^2$.

## C.3  $\alpha$-REGULARITY VS CONTEXT DIVERSITY

In recent years, there has been significant attention on the optimality of the Greedy algorithm in 1-objective bandit problems (Bastani et al., 2021; Kannan et al., 2018; Raghavan et al., 2018; Hao et al., 2020). A common theme among these studies is assuming that the feature vector follows a continuous distribution that satisfies specific diversity conditions. MORR-Greedy always achieve optimal regret under any form of context diversity assumptions which can drive $\lambda_{\min}(\mathbb{E}[x(t)x(t)^\top | \mathcal{H}_{t-1}]) \geq \lambda_0$. Given the results in the single-objective setting, this is not surprising at all.

The $\gamma$-regularity condition is fundamentally different from context diversity condition. Naturally, in the single-objective setting, we cannot guarantee that the greedy algorithm will perform well under the $\gamma$-regularity condition. However, when the number of objectives is sufficiently large, the intersection of the multi-objective setting and the $\gamma$-regularity condition leads to free exploration, resulting in optimal regret. The following example highlight cases where the assumption of context diversity is not met, yet the regularity condition remains valid, enabling the MORR-Greedy optimal performance in many-objective bandit problems.

**Example 1** (Containing Fixed Arms) Imagine a situation where one feature vector is a continuous variable while the other arms are fixed. For example, let $x_1(t)$ be uniformly distributed over $\mathbb{B}^d$ while $x_2(t) = x_2, \ldots, x_K(t) = x_K$ are fixed at some points in $\mathbb{S}^{d-1}$. By Lemma C.1, $P_{\chi}(t)$ satisfies $\gamma$-regularity for all $\gamma \in (0, 1)$. However, it is easy to see that diversity is not satisfied because $\lambda_{\min}(\mathbb{E}[x(t)x(t)^\top | \hat{\theta}(t) = x_2]) = \lambda_{\min}(x_2 x_2^\top) = 0$.

**Example 2** (Low-Randomness Distribution) Consider a scenario where the feature vectors are drawn from a finite set of discrete points. Despite the lack of diversity, if these points are strategically chosen to cover $\mathbb{S}^{d-1}$ adequately, the regularity condition can still be satisfied. For example, suppose there is a set of points $P = \{a_1, a_2, \ldots, a_N\}$ that contains $\sqrt{1 - \gamma^2}$-net of $\mathbb{S}^{d-1}$. Assume that $x_1(t)$ be chosen uniformly from the $N_1 < d$ points in $P$ with the largest first coordinates, and other arms $x_2(t), \ldots, x_K(t)$ be chosen from the remaining points. Obviously, $P_{\chi}(t)$ satisfies $\gamma$-regularity with $q_\gamma \geq \frac{1}{N}$. However, greedy selection by the vector $(1, 0, \ldots, 0)$ should be $x_1(t)$ and there are only $N_1 < d$ candidates that can be $x_1(t)$. Therefore, context diversity does not hold in this scenario.

Although $\gamma$-regularity encompasses cases where context diversity is not covered, there is no inclusion relationship between the two conditions. Here is an example where regularity does not hold, but context diversity does.

**Example 3** (Proper Support) Consider a case where 1 is given as the upper bound of the $l_2$ norm of feature vectors, but the actual support of feature vectors is smaller. For instance, if $x_i(t)$ follows a

uniform distribution over $\mathbb{B}_{1/2}^d$ for all $i \in [K]$ and $t \in [T]$, then context diversity still holds (Bastani et al. (2021)), but $\gamma$-regularity does not hold for $\gamma > 1/2$.

# D  ANOTHER PROPOSED ALGORITHM

## D.1  MULTI-OBJECTIVE RANDOM OBJECTIVE – GREEDY ALGORITHM

The following algorithm is the `MORO-Greedy` algorithm. In each round, it randomly selects one objective and chooses the arm greedily based on this objective. The algorithm takes as input the probabilities for selecting each objective, which can be uniformly set to $\frac{1}{M}$ if no specific information is available. Similar to the MORR-Greedy algorithm, this algorithm utilizes the initial $\beta_1, \ldots, \beta_M$ values until $\lambda_{\min}(V_{t-1}) > \lambda$ for a threshold $\lambda$, and then uses the OLS estimators $\hat{\theta}_m(t)$ of $\theta_m^*$ in each round $t$.

---

**Algorithm D.1** Multi-Objective Random Objective – Greedy Algorithm (`MORO-Greedy`)

---

**Require:** $T, \lambda, (w_1, \ldots, w_M)$ {Parameters: Total rounds $T$, minimum eigenvalue threshold $\lambda$, the distribution of objectives $(w_1, \ldots, w_M)$}

1: Initialize $V_0 \leftarrow 0 \times I_d$, and $\beta_1, \ldots, \beta_M \in \mathbb{R}^d$
2: **for** $t = 1$ **to** $T$ **do**
3:     Randomly select $m \in [M]$ from the distribution $(w_1, \ldots, w_M)$.
4:     **if** $\lambda_{\min}(V_{t-1}) > \lambda$ **then**
5:         Update the OLS estimators $\hat{\theta}_1(t), \ldots, \hat{\theta}_M(t)$
6:         Select action $a(t) \in \arg\max_{i \in [K]} x_i^\top \hat{\theta}_m(t)$
7:     **else**
8:         Select action $a(t) \in \arg\max_{i \in [K]} x_i^\top \beta_m$
9:     **end if**
10:   Observe the reward vector $y(t) = \left(y_{a(t),1}(t), \ldots, y_{a(t),M}(t)\right)$
11:   Update $V_t \leftarrow V_{t-1} + x(t)x(t)^\top$
12: **end for**

---

The `MORO-Greedy` algorithm can be viewed as operating a greedy algorithm in a multi-objective setting where the dominant objective changes with each round randomly. Specifically, in a scenario where the dominant objective varies across users' preferences and no objective has a zero probability of being dominant, the result of this section shows that simply applying a greedy algorithm to the dominant objective could be an optimal strategy.

## D.2  ANALYSIS OF MORO-GREEDY ALGORITHM

The following lemma demonstrates that objective diversity in `MORO-Greedy` leads to context diversity even when the features are fixed. The expectation of the lemma below arises not from the randomness of the contexts, but rather from the randomness associated with the selection of the algorithm's objectives.

**Lemma D.1.** *Suppose Assumptions 1, 2, and 3 hold. Assume the OLS estimator satisfies $||\hat{\theta}_m(s)) - \theta_m^*|| \leq \alpha_0$, for all $m \in [M]$ and $s \geq T_0 + 1$. Then, for $s \geq T_0 + 1$, the arms selected by Algorithm 2 satisfies*

$$\lambda_{\min}(\mathbb{E}[x(s)x(s)^\top | \mathcal{H}_{s-1}]) \geq q_0 \min_{m \in [M]} (w_m) C_0,$$

*where $C_0 := \lambda_0 - 2\sqrt{2\left(1 + \alpha_0\sqrt{1 - \gamma_0^2} - \gamma_0\sqrt{1 - \alpha_0^2}\right)}$.*

*Proof of Lemma D.1.* For $s \geq T_0 + 1$, let $E_m(s)$ be the event that the objective $m$ is a target objective and define the near optimal zone $R_m(s) := \{x \in \mathbb{B}^d \mid x^\top \frac{\hat{\theta}_m(s)}{||\hat{\theta}_m(s)||} \geq \gamma_0\}$ on round $s$. Then,

$$
\mathbb{E}[x(s)x(s)^\top|\mathcal{H}_{s-1}]
$$

$$
= \sum_{m=1}^{M} \mathbb{E}[x(s)x(s)^\top|E_m(s), \mathcal{H}_{s-1}]\mathbb{P}[E_m(s)|\mathcal{H}_{s-1}]
$$

$$
\succeq \sum_{m=1}^{M} w_m \mathbb{E}[x(s)x(s)^\top \mid x(s) \in R_m(s), E_m(s), \mathcal{H}_{s-1}] \, \mathbb{P}[x(s) \in R_m(s)|E_m(s), \mathcal{H}_{s-1}]
$$

$$
= \sum_{m=1}^{M} w_m \mathbb{E}[x(s)x(s)^\top \mid x(s) \in R_m(s), E_m(s), \mathcal{H}_{s-1}]\mathbb{P}[\exists i \in [K], x_i \in R_m(s)|E_m(s), \mathcal{H}_{s-1}]
$$

$$
\succeq q_0 \min_{m \in [M]} (w_m) \sum_{m=1}^{M} \mathbb{E}[x(s)x(s)^\top \mid x(s) \in R_m(s), E_m(s), \mathcal{H}_{s-1}]
$$

$$
\succeq q_0 \min_{m \in [M]} (w_m) \sum_{m=1}^{M} \mathbf{x}_m(s)\mathbf{x}(s)^\top,
$$

where $\mathbf{x}_m(s) := \mathbb{E}[x(s)| x(s) \in R_m(s), E_m(s), \mathcal{H}_{s-1}]$. The second inequality from the bottom holds due to the fact that the existence of a near-optimal arm is independent of the choice of the target objective or the history $\mathcal{H}_{t-1}$, as established by Assumption 3. The final line is validated by Lemma H.4.

Since $R_m(s)$ is a convex set for all $m \in [M]$ and $s \geq T_0 + 1$, $\mathbf{x}_m(s)$ should be inside $R_m(s)$, which allows us to apply Lemma B.1 to above inequality by

$$
\lambda_{\min}(\mathbb{E}[x(s)x(s)^\top|\mathcal{H}_{s-1}]) \geq q_0 \min_{m \in [M]} (w_m) \lambda_{\min}(\sum_{m=1}^{M} \mathbf{x}_m(s)\mathbf{x}_m(s)^\top)
$$

$$
\geq q_0 \min_{m \in [M]} (w_m) \left( \lambda_0 - 2\sqrt{2\left(1 + \alpha_0\sqrt{1-\gamma_0^2} - \gamma_0\sqrt{1-\alpha_0^2}\right)} \right).
$$

∎

We can prove the minimum eigenvalue of the gram matrix increases linearly with respect to $t$ with Lemma D.1. This leads that the `MORO-Greedy` algorithm can have the same scale of Pareto regret bound and objective fairness as in `MORR-Greedy`.

**Corollary D.1** (Pareto Regret of `MORO-Greedy`). *Given Assumption 1, 2, and 3, the `MORO-Greedy` algorithm has a regret bound of $\widetilde{\mathcal{O}}(\sqrt{dT})$.*

**Corollary D.2** (Objective Fairness of `MORO-Greedy`). *Given Assumption 1, 2, and 3, the `MORO-Greedy` algorithm satisfies the objective fairness.*

# E   RELEASING BOUNDEDNESS ASSUMPTION

In this section, we explain how to release the boundedness assumption, Assumption 1. In conclusion, we can obtain results of the same scale as Theorems 1 and 2 for any arbitrary bound on feature vector. $||x_i||_2 \leq x_{\max}$ and $l \leq \theta_m^* \leq L$ for all $m \in [M]$. For clarity, we will separately discuss how to release the $l_2$ norm bounds of the feature vector and the objective parameters in Appendix E.1 and E.2, respectively. However, It is important to note that there is no issue in applying the same argument even when the bound on the feature vectors and the bound on the objective parameters are released simultaneously. We present how to release the boundedness assumption in fixed features setting, but the same reasoning can be applied to the case of stochastic contexts.

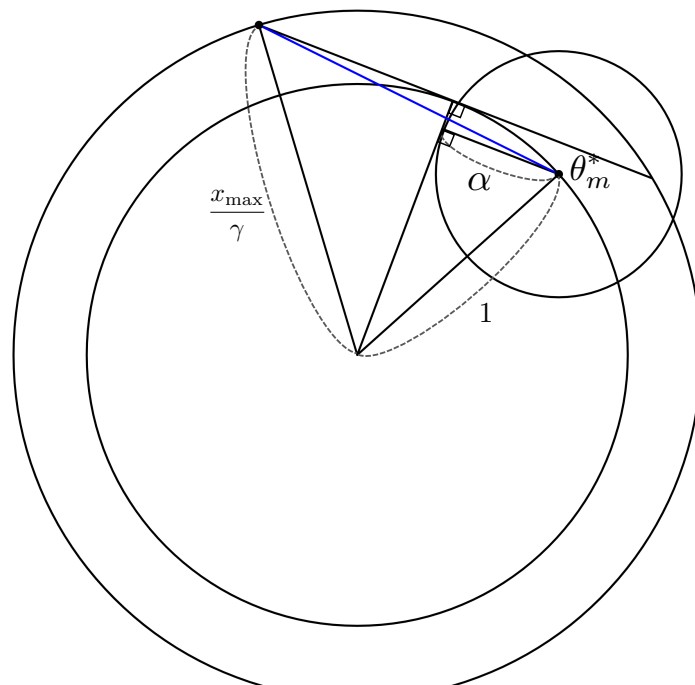

Figure E.1: The interior of the circle with radius $\frac{x_{\max}}{\gamma}$ represents the region where $\frac{x}{\gamma}$ may exist in $\mathbb{R}^d$, while that of the smallest circle indicates the region where $\hat{\theta}_m(s)$ may exist. Then, the blue line illustrates the case when $\frac{x}{\gamma}$ is farthest from the $\theta_m^*$.

### E.1 RELEASING BOUND ON FEATURE VECTORS

We demonstrate how the minimum eigenvalue of the Gram matrix can increase linearly when the $l_2$ norm of the feature vectors is bounded by an arbitrary upper bound $x_{\max}$. The $\gamma$-regularity assumption is related to the scale of the feature, and thus, when we modify the bound of the boundedness assumption, the $\gamma$-regularity assumption also change correspondingly.

**Assumption E.1** (Boundedness). $\forall i \in [K]$, $||x_i||_2 \leq x_{\max}$, and $\forall m \in [M]$, $||\theta_m^*||_2 = 1$.

**Assumption E.2** ($\gamma$-Regularity). *We assume* $\{x_1, \ldots, x_K\}$ *satisfies* $\gamma_0$*-regular with* $\gamma_0 < \frac{x_{\max}}{\lambda_0}\sqrt{2\sqrt{1 + \lambda_0^2} - 2}$.

The following lemma is the key of the releasing process.

**Lemma E.1.** *Given Assumptions E.1, assume the OLS estimator satisfies* $||\hat{\theta}_m(s) - \theta_m^*|| \leq \alpha$, *for* $m \in [M]$ *and* $s \geq T_0 + 1$. *If* $x \in \mathbb{B}_{x_{\max}}^d$ *satisfies* $x^\top \frac{\hat{\theta}_m(s)}{||\hat{\theta}_m(s)||} \leq \gamma$, *then the distance between* $\frac{x}{\gamma}$ *and* $\theta_m^*$ *is bounded by*

$$\left\|\theta_m^* - \frac{x}{\gamma}\right\|_2 \leq \sqrt{1 + (\frac{x_{\max}}{\gamma})^2 + 2\alpha\sqrt{(\frac{x_{\max}}{\gamma})^2 - 1} - 2\sqrt{1 - \alpha^2}}.$$

*Proof of Lemma E.1.* Consider the case when $\frac{x}{\gamma}$ is the farthest from $\theta_m^*$. As we easily can see from Figure E.1,

$$\left\|\theta_m^* - \frac{x}{\gamma}\right\|_2^2 \leq \left(\alpha + \sqrt{(\frac{x_{\max}}{\gamma})^2 - 1}\right)^2 + (1 - \sqrt{1 - \alpha^2})^2$$

$$= 1 + (\frac{x_{\max}}{\gamma})^2 + 2\alpha\sqrt{(\frac{x_{\max}}{\gamma})^2 - 1} - 2\sqrt{1 - \alpha^2}.$$

■

**Corollary E.1.** *Suppose Assumptions E.1, 2, and E.2 hold. Assume the OLS estimator satisfies* $||\hat{\theta}_m(s) - \theta_m^*|| \leq \alpha_0$*, for all* $m \in [M]$ *and* $s \geq T_0 + 1$*. Then, the selected arms for a single cycle* $s = t_0,\ t_0 + 1,\ \dots,\ t_0 + M - 1\ (t_0 > T_0)$ *by Algorithm1 satisfies*

$$\lambda_{\min}\left(\sum_{s=t_0}^{t_0+M-1} x(s)x(s)^\top\right) \geq \left(\lambda_0\gamma_0^2 - 2x_{\max}\sqrt{\gamma_0^2 + x_{\max}^2} + 2\alpha_0\sqrt{x_{\max}^2 - \gamma_0^2} - 2\gamma_0^2\sqrt{1 - \alpha_0^2}\right) M.$$

The above corollary means that even when Assumptions 1 and 3 are replaced by Assumptions E.1 and E.2, respectively, we can still obtain a regret bound that differs by at most a constant factor. Furthermore, using the same argument as before, we can also verify the objective fairness with replaced assumptions.

## E.2 RELEASING BOUND ON OBJECTIVE PARAMETERS

In this section, we present how to handle objective parameters with varying $l_2$ norm sizes. The $\gamma$-regularity assumption is related to the scale of the objectives either, the $\gamma$-regularity assumption is modified again correspondingly.

**Assumption E.3** (Boundedness). $\forall i \in [K]$, $||x_i||_2 \leq 1$, and $\forall m \in [M]$, $l \leq ||\theta_m^*||_2 \leq L$.

**Assumption E.4** ($\gamma$-Regularity). *We assume* $\{x_1, \dots, x_K\}$ *satisfies* $\gamma_0$*-regular with* $\gamma_0 < 1 - \frac{\lambda_0^2}{8L^4}$.

The following lemma is the key of the releasing process.

**Lemma E.2.** *Given Assumptions E.3, assume the OLS estimator satisfies* $||\hat{\theta}_m(s) - \theta_m^*|| \leq \alpha$*, for* $m \in [M]$ *and* $s \geq T_0 + 1$*. If* $x \in \mathbb{B}^d$ *satisfies* $x^\top \frac{\hat{\theta}_m(s)}{||\hat{\theta}_m(s)||} \leq \gamma$*, then the distance between* $x$ *and* $\frac{\theta_m^*}{||\theta_m^*||_2}$ *is bounded by*

$$\left\|\frac{\theta_m^*}{||\theta_m^*||_2} - x\right\|_2 \leq \sqrt{2(1 + (\frac{\alpha}{l})\sqrt{1 - \gamma^2} - \gamma\sqrt{1 - (\frac{\alpha}{l})^2})}.$$

*Proof of Lemma E.2.* Consider the case when $x$ is the farthest from $\frac{\theta_m^*}{||\theta_m^*||_2}$. As we easily can see from Figure E.2, we can obtain the following result from Lemma A.1by replacing $\alpha$ by $\frac{\alpha}{l}$

$$\left\|\frac{\theta_m^*}{||\theta_m^*||_2} - x\right\|_2 \leq \sqrt{2(1 + (\frac{\alpha}{l})\sqrt{1 - \gamma^2} - \gamma\sqrt{1 - (\frac{\alpha}{l})^2})}.$$

■

**Corollary E.2.** *Suppose Assumptions E.3, 2, and E.4 hold. Assume the OLS estimator satisfies* $||\hat{\theta}_m(s) - \theta_m^*|| \leq \alpha_0$*, for all* $m \in [M]$ *and* $s \geq T_0 + 1$*. Then, the selected arms for a single cycle* $s = t_0,\ t_0 + 1,\ \dots,\ t_0 + M - 1\ (t_0 > T_0)$ *by Algorithm1 satisfies*

$$\lambda_{\min}\left(\sum_{s=t_0}^{t_0+M-1} x(s)x(s)^\top\right) \geq \left(\frac{\lambda_0}{L^2} - 2\sqrt{2\left(1 + (\frac{\alpha_0}{l})\sqrt{1 - \gamma_0^2} - \gamma_0\sqrt{1 - (\frac{\alpha_0}{l})^2}\right)}\right) M.$$

The corollary can be derived from Lemma E.2 and $\lambda_{\min}\left(\frac{1}{M}\sum_{m=1}^{M}\left(\frac{\theta_m^*}{||\theta_m^*||_2}\right)\left(\frac{\theta_m^*}{||\theta_m^*||_2}\right)^\top\right) \geq \frac{\lambda_0}{L^2}$.

Therefore, we can still obtain a regret bound that differs by at most a constant factor and the objective fairness criterion with Assumption E.3 and Assumption E.4.

## F OBJECTIVE DIVERSITY ON FEATURE VECTOR SPACE

Until now, we have conducted an analysis under the assumption that the feature vectors span $\mathbb{R}^d$. Although this assumption was not explicitly stated, it can be derived from objective diversity(Assumption 2) and $\gamma$-regularity(Assumption 3). In this chapter, we present a sufficient condition under which MORR-Greedy performs well when the feature vectors do not span $\mathbb{R}^d$ and explain how this leads to regret bounds and objective fairness.

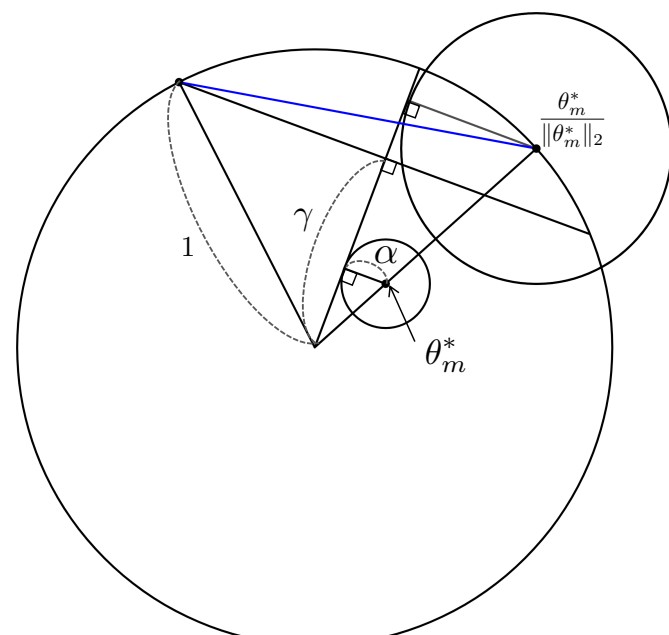

Figure E.2: The larger circle represents the unit sphere in $\mathbb{R}^d$ while the interior of the smallest circle indicates the region where $\hat{\theta}_m(s)$ may exist. Then, the blue line illustrates the case when $x$ is farthest from the $\frac{\theta_m^*}{||\theta_m^*||_2}$

.

**Additional Notations.** We denote the spanning space of feature vectors $x_1, \ldots, x_K$ by $S_x$ and the orthogonal complement of $S_x$ by $S_x^\perp$. We use $\pi_S(v) : \mathbb{R}^d \to S_x$ as a projection map onto $S_x$.

**Intuition.** It is evident that any bandit algorithm cannot obtain information about the true objective parameters in the direction of $S^\perp$ while interacting with feature vectors $x_1, \ldots, x_K$. In other words, during the process of estimating the objective parameters, no estimator can converge to the true parameters in the direction of space $S^\perp$. Interestingly, from the perspective of regret and optimality, this poses no significant issue. This can be expressed mathematically as for any pair of arms $i, j \in [K]$ and $m \in [M]$,

$$x_i^\top \theta_m^* - x_j^\top \theta_m^* = x_i^\top (\pi_S(\theta_m^*)) - x_j^\top (\pi_S(\theta_m^*)).$$

The above equation explains why regret and optimality are determined solely by the projection vector of the objective parameters onto $S$.

Before we begin the analysis, it is important to address the OLS estimator used in `MORR-Greedy`. When the feature vectors do not span $\mathbb{R}^d$, a unique least squares solution no longer exists, and the closed form cannot be utilized. Therefore, we use an arbitrary solution $\hat{\theta}_m(t)$ of $\left( \sum_{s=1}^{t-1} x(s)x(s)^\top \right) \theta = \sum_{s=1}^{t-1} x(s)y_j(s)$ for each round $t$.

The following are the revised versions of Assumptions 1, 2, and 3, adapted to the feature vector space.

**Assumption F.1** (Boundedness). $\forall i \in [K]$, $||x_i||_2 \leq 1$, and $\forall m \in [M]$, $||\pi_s(\theta_m^*)||_2 = 1$.

Once again, the above assumption is intended for a clear analysis. The analyses conducted in this section can be also extended to arbitrary bounds $||x_i||_2 \leq x_{\max}$ and $l \leq \pi_s(\theta_m^*) \leq L$ for all $m \in [M]$ by the same process in Appendix E.

**Assumption F.2** (Objective Diversity on $S_x$). *We assume* $\theta_1^*, \ldots, \theta_M^*$ *span* $S_x$.

In the following analysis, we define $\lambda_1 := \min_{||\beta||=1, \beta \in S_x} \left( \frac{1}{M} \sum_{m=1}^{M} \langle \beta, \theta_m^* \rangle^2 \right)$, the degree of the objective diversity on $S_x$. Then, given Assumption F.2, $\lambda_1$ is always positive.

**Definition 8** ($\gamma$-Regular Condition on $S_x$). *For fixed $\gamma \in (0, 1]$, we say that the set of feature vectors $\{x_1, \ldots, x_K\}$ satisfies $\gamma$-regular condition when there exists $\alpha > 0$ that satisfies*

$$\forall \beta \in \mathbb{B}_\alpha(\pi_S(\theta_1^*)) \cup \ldots \cup \mathbb{B}_\alpha(\pi_S(\theta_M^*)), \quad \exists i \in [K], \quad x_i^\top \frac{\beta}{||\beta||_2} \geq \gamma. \tag{F.1}$$

**Assumption F.3** ($\gamma$-Regularity on $S_x$). *We assume $\{x_1, \ldots, x_K\}$ satisfies $\gamma_0$-regular with $\gamma_0 > 1 - \frac{\lambda_1^2}{8}$.*

Once again, in the following analysis, $\alpha_0$ denotes the value of $\alpha$ that holds the condition (2). If $\alpha_0$ is greater than $\sqrt{\frac{\lambda_1^2}{4} - \frac{\lambda_1^4}{64}} \, \gamma_0 - \left(1 - \frac{\lambda_1^2}{8}\right)\sqrt{1 - \gamma_0^2}$, then set $\alpha_0$ slightly less than this value.

The only question is how to construct an $l_2$ bound on $\pi_S(\hat{\theta}_m(s)) - \pi_S(\theta_m^*)$ without utilizing the minimum eigenvalue of the Gram matrix, which is zero when $S_x \subsetneq \mathbb{R}^d$. The key idea is that we can use $\min_{||\beta||=1, \, \beta \in S_x} \left(\sum_{s=1}^{t-1} \langle \beta, \, x(s) \rangle^2\right)$ to fulfill the role previously played by the minimum eigenvalue. We present 2 Lemmas, Lemma F.1 and Lemma F.2, to explain the idea. First, The following demonstrates the linear growth of $\min_{||\beta||=1, \, \beta \in S_x} \left(\sum_{s=1}^{t-1} \langle \beta, \, x(s) \rangle^2\right)$.

**Lemma F.1.** *Suppose Assumptions F.1, F.2, and F.3 hold. Assume a least square solution $\hat{\theta}_m(s)$ satisfies $||\pi_S(\hat{\theta}_m(s)) - \pi_S(\theta_m^*)|| \leq \alpha_0$, for all $m \in [M]$ and $s \geq T_0 + 1$. Then, the selected arms for a single cycle $s = t_0, \, t_0 + 1, \, \ldots, \, t_0 + M - 1$ ($t_0 > T_0$) by Algorithm1 satisfies*

$$\min_{||\beta||=1, \, \beta \in S_x} \left(\sum_{s=t_0}^{t_0+M-1} \langle \beta, \, x(s) \rangle^2\right) \geq \left(\lambda_1 - 2\sqrt{2\left(1 + \alpha_0\sqrt{1 - \gamma_0^2} - \gamma_0\sqrt{1 - \alpha^2}\right)}\right) M.$$

*Proof of Lemma F.1.* Since the greedy selection of $\hat{\theta}_m(s)$ is equal to that of $\pi_S(\hat{\theta}_m(s))$, for the same reason as Lemma A.1 we can get $||x(s) - \pi_S(\theta_{m(s)}^*)||_2 \leq \sqrt{2\left(1 + \alpha_0\sqrt{1 - \gamma_0^2} - \gamma_0\sqrt{1 - \alpha_0^2}\right)}$, where $j_s$ is the target objective for round $s$.

Then, for any unit vector $\beta \in S_x$,

$$\beta^\top \left(\sum_{s=t_0}^{t_0+M-1} x(s)x(s)^\top\right)\beta$$

$$= \sum_{s=t_0}^{t_0+M-1} \left\{\left\langle \beta, \, \pi_S(\theta_{m(s)}^*)\right\rangle^2 + \left\langle \beta, \, x(s) - \pi_S(\theta_{m(s)}^*)\right\rangle^2 + 2\left\langle \beta, \, \pi_S(\theta_{m(s)}^*)\right\rangle\left\langle \beta, \, x(s) - \pi_S(\theta_{m(s)}^*)\right\rangle\right\}$$

$$\geq M\lambda_1 - 2\sqrt{2\left(1 + \alpha_0\sqrt{1 - \gamma_0^2} - \gamma_0\sqrt{1 - \alpha_0^2}\right)}M.$$

$\blacksquare$

The next lemma shows how to derive $l_2$ bound on $\pi_S(\hat{\theta}_m(s)) - \pi_S(\theta_m^*)$ with $\min_{||\beta||=1, \, \beta \in S_x} \left(\sum_{s=1}^{t-1} \langle \beta, \, x(s) \rangle^2\right)$.

**Lemma F.2.** *For all $m \in [M]$ and $t \geq 1$, any least square solution $\hat{\theta}_m(t)$ of $\left(\sum_{s=1}^{t-1} x(s)x(s)^\top\right)\theta = \sum_{s=1}^{t-1} x(s)y_{a(s),m}(s)$ satisfies*

$$\left|\left|\pi_S(\hat{\theta}_m(s)) - \pi_S(\theta_m^*)\right|\right|_2 \leq \frac{||\sum_{s=1}^{t-1} x(s)\eta_{a(s),m}(s)||_2}{\min_{||\beta||=1, \, \beta \in S_x} \left(\sum_{s=1}^{t-1} \langle \beta, \, x(s) \rangle^2\right)}.$$

*Proof of Lemma F.2.* From the definition of $\hat{\theta}_m(t)$, we have

$$\left(\sum_{s=1}^{t-1} x(s)x(s)^\top\right)(\hat{\theta}_m(t) - \theta_m^*) = \sum_{s=1}^{t-1} x(s)\eta_{a(s),m}(s).$$

Since the row space of $\left( \sum_{s=1}^{t-1} x(s)x(s)^\top \right)$ is in $S$,

$$\left\| \sum_{s=1}^{t-1} x(s)\eta_{a(s),m}(s) \right\|_2 = \left\| \left( \sum_{s=1}^{t-1} x(s)x(s)^\top \right) \left( \pi_S(\hat{\theta}_m(s)) - \pi_S(\theta_m^*) \right) \right\|_2$$

$$\geq \min_{||\beta||=1,\ \beta \in S_x} \left( \beta^\top \left( \sum_{s=1}^{t-1} x(s)x(s)^\top \right)\beta \right) \left\| \pi_S(\hat{\theta}_m(s)) - \pi_S(\theta_m^*) \right\|_2 .$$

The last inequality holds by Lemma H.5.

∎

With above two lemmas, we can obtain the same regret bound and objective fairness as in Theorem 1 and 2.

## G EXPERIMENTS

### G.1 EMPIRICAL VALIDATION OF PERFORMANCE OF MORR-GREEDY

We compare the empirical Pareto regret of MORR-Greedy in a linear bandit setting with other multi-objective algorithms. We experiment with a linear bandit, $y_m(t) = \mathcal{N}(x_i^T \theta_m^*, 0.1^2)$ for all $i \in [K]$ and $m \in [M]$. For each problem instance, $M$ objective parameters are selected randomly uniformly from the positive part of $\mathbb{S}^{d-1}$ and then $K(> 2M)$ feature vectors are drawn from $\mathbb{B}^d$. In the case of fixed arms, to ensure a certain degree of regularity, the first $M$ feature vectors are drawn from the multivariate normal distribution with the true objective parameter as the mean and a covariance matrix of $0.1 I_d$. Subsequently, we use the vectors were scaled to ensure that their magnitudes fall within the range of $(3/4, 1)$. The other $K - M$ vectors are drawn uniformly at random from $\mathbb{B}^d$. Among $M$ of $K - M$ vectors are scaled to have length longer than $3/4$, while the rest of them to have their magnitudes smaller than $3/4$. Limiting the magnitudes of the feature vectors prevents meaningless results caused by excessively large Pareto fronts. For the experiment with varying arms, contexts are drawn from uniform distribution on $\mathbb{B}^d$. Our results are averaged over 10 different instances for each $(d, K, M)$-combination, and we conduct 10 reputations for 1 problem instance.

We conduct experiments on MORR-Greedy and the two base lines, P-UCB (Drugan & Nowe, 2013) and MOGLM-UCB (Lu et al., 2019) with tuned parameters for contextual algorithms. We evaluated the performance of each algorithm in both cases of fixed arms and stochastic arms. When playing with stochastic arms, the feature vectors are drawn the uniform distribution on $\mathbb{B}^d$ and only contextual algorithms MORR-Greedy and MOGLM-UCB are compared. For MOGLM-UCB, we use the confidence width $\gamma_t = c \log \frac{\det(Z_t)}{\det(Z_1)}$ where $Z_t = I_d + \frac{1}{2} \sum_{s=1}^{t} x(s)x(s)^\top$ for $c = 1$ and 0.1, as used in Lu et al. (2019). For MORR-Greedy, we use the parameter $\lambda = 1$, 0.1, and 0.01.

Results are shown in Figure G.1 and Figure G.2. In both case, MORR-Greedy outperforms MOGLM-UCB and MORR-Greedy with the smallest $\lambda$ was the best. This shows that when the objectives are diverse in multi-objective bandit, short exploration is enough to make the algorithm perform well.

### G.2 EMPIRICAL VALIDATION OF OBJECTIVE FAIRNESS OF MORR-GREEDY

We empirically confirmed that our algorithm fairly selects near-optimal arms for each objective. To validate this, we empirically calculated the objective fairness index $p_{\epsilon,T}$ at two different $\epsilon$ levels: 0.1 and 0.05, for each simulation. The experimental setup was identical to that in Setting G.1.

As shown in Figure G.3 and Figure G.4, in both the fixed arm and stochastic arm cases, MORR-Greedy selected the $\epsilon$-optimal arms for all objectives at proportions of at least $\frac{1}{M}$ for both $\epsilon$ levels (0.1 and 0.05) in most cases, and the influence of the difference in $K$ was not significant. The proportion may exceed $\frac{1}{M}$ because the $\epsilon$-optimal arm sets for the different objectives may overlap. (In this experiment, arms were drawn from $\mathbb{B}^d$, and objective parameters were drawn only from the positive part of $\mathbb{S}^{d-1}$, resulting in frequent overlap of the $\epsilon$-optimal arm sets for the objectives.)

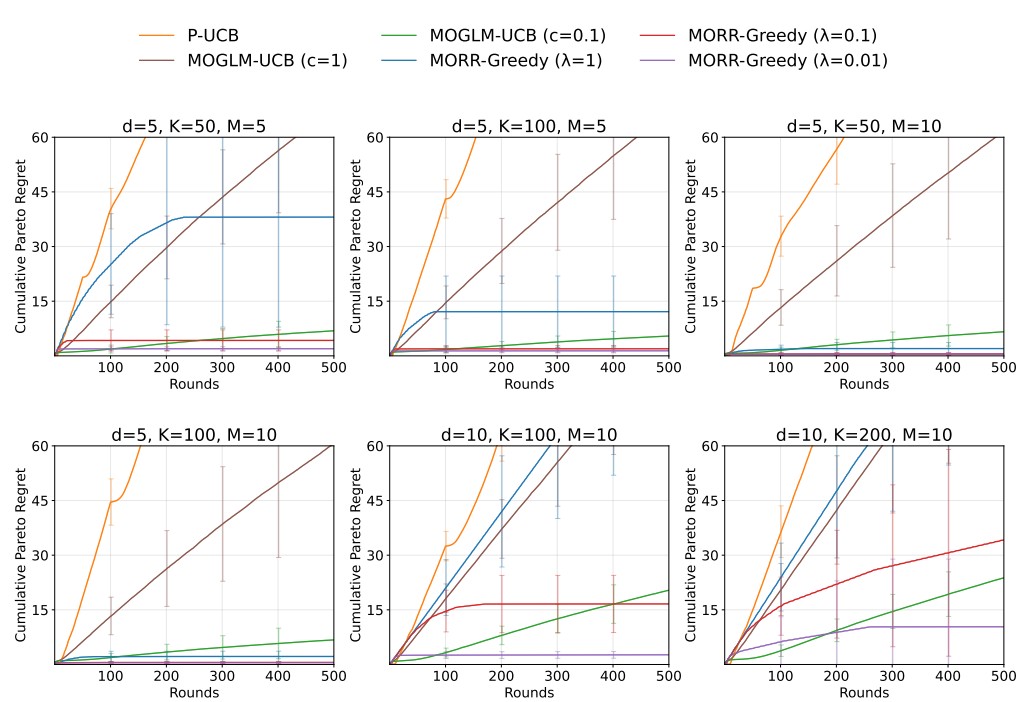

Figure G.1: Evaluation of multi-objective bandit algorithms playing with fixed arms for various $(d, K, M)$

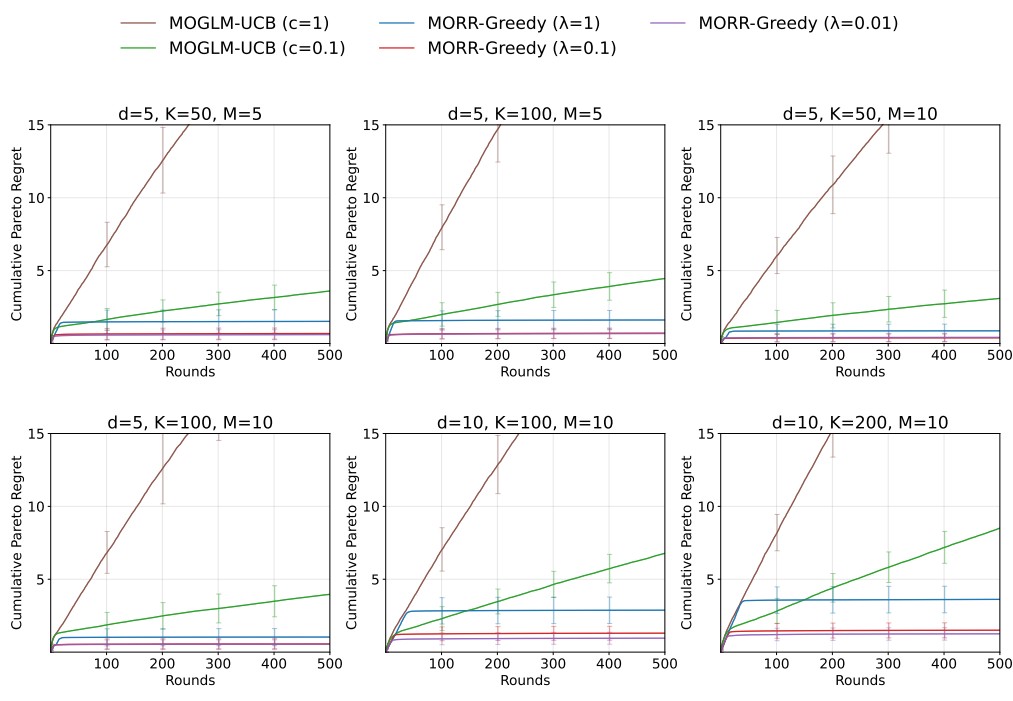

Figure G.2: Evaluation of multi-objective bandit algorithms playing with stochastic arms for various $(d, K, M)$

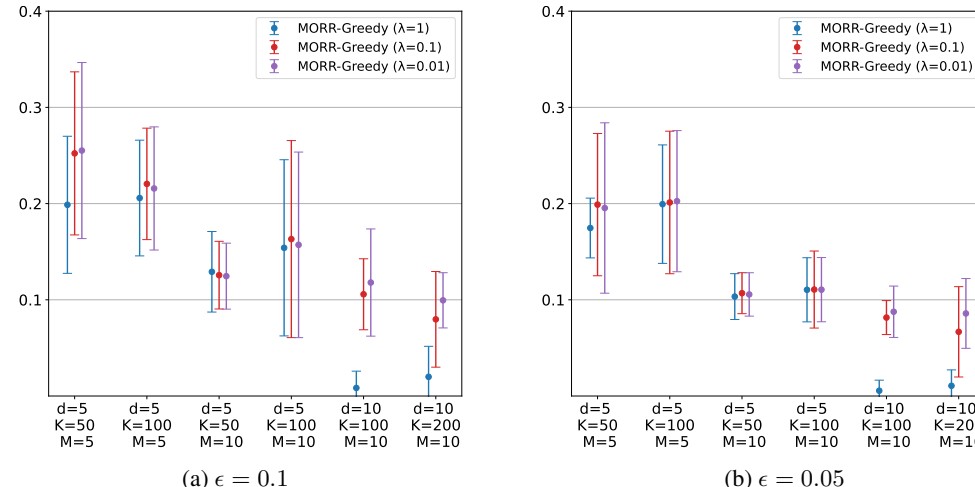

(a) $\epsilon = 0.1$                 (b) $\epsilon = 0.05$

Figure G.3: Evaluation of the objective fairness index of the MORR-Greedy algorithm playing with fixed arms. The $y$-axis represents the minimum proportion of rounds the $\epsilon$-optimal arm is selected for each objective.

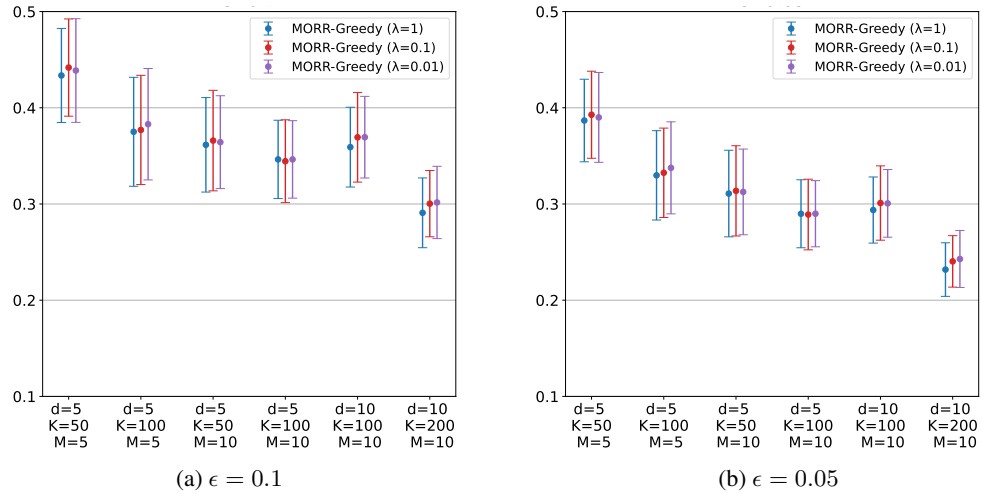

(a) $\epsilon = 0.1$                 (b) $\epsilon = 0.05$

Figure G.4: Evaluation of the objective fairness index of the MORR-Greedy algorithm playing with stochastic arms. The $y$-axis represents the minimum proportion of rounds the $\epsilon$-optimal arm is selected for each objective.

### G.3 EXPERIMENT ON THE INFLUENCE OF INITIAL OBJECTIVE PARAMETERS

We conducted experiments to investigate the impact of the initial objective parameters on the algorithm's performance. In the case of fixed arms, once $\beta_1, \ldots, \beta_M$ are set, the arms that is optimized to $\beta_1, \ldots, \beta_M$ will be selected during the exploration process. Thus, when the arms are invariant, instead of explicitly setting $\beta_1, \ldots, \beta_M$, we only have to select $M$ arms that span $\mathbb{R}^d$. Therefore, we measured the algorithm's performance for various beta combinations in a stochastic context setting. We test the influence of $\beta_1, \ldots, \beta_M$ on four combinations of $(d, K, M) = (5, 50, 5), (5, 100, 5), (10, 50, 10), (10, 100, 10)$. For each $(d, K, M)$-combination, our results are averaged over 10 reputations for each 10 different problem instances. For each simulation, we considered three different sets of $\beta_1, \ldots, \beta_M$ based on diversity. When $M = 5$, with the standard basis of $\mathbb{R}^d$ denoted as $\{e_1^{(5)}, \ldots, e_5^{(5)}\}$, we considered the most diverse beta set as the standard basis, the moderately diverse beta set as $\left\{ \frac{e_1^{(5)} + e_2^{(5)}}{\sqrt{2}}, \frac{e_2^{(5)} + e_3^{(5)}}{\sqrt{2}}, \ldots, \frac{e_5^{(5)} + e_1^{(5)}}{\sqrt{2}} \right\}$, and the least diverse beta set as

$\left\{ \frac{e_1^{(5)}+e_2^{(5)}+e_3^{(5)}}{\sqrt{3}}, \frac{e_2^{(5)}+e_3^{(5)}+e_4^{(5)}}{\sqrt{3}}, \ldots, \frac{e_5^{(5)}+e_1^{(5)}+e_2^{(5)}}{\sqrt{3}} \right\}$. When $M = 10$, with the standard basis of $\mathbb{R}^d$ denoted as $\{e_1^{(10)}, \ldots, e_{10}^{(10)}\}$, we considered the most diverse beta set as the standard basis, the moderately diverse beta set as $\left\{ \frac{e_1^{(10)}+e_2^{(10)}+e_3^{(10)}}{\sqrt{3}}, \frac{e_2^{(10)}+e_3^{(10)}+e_4^{(10)}}{\sqrt{3}}, \ldots, \frac{e_{10}^{(10)}+e_1^{(10)}+e_2^{(10)}}{\sqrt{3}} \right\}$, and the least diverse beta set as $\left\{ \frac{e_1^{(10)}+e_2^{(10)}+e_3^{(10)}+e_4^{(10)}+e_5^{(10)}+e_6^{(10)}}{\sqrt{6}}, \ldots, \frac{e_{10}^{(10)}+e_1^{(10)}+e_2^{(10)}+e_3^{(10)}+e_4^{(10)}+e_5^{(10)}}{\sqrt{6}} \right\}$.

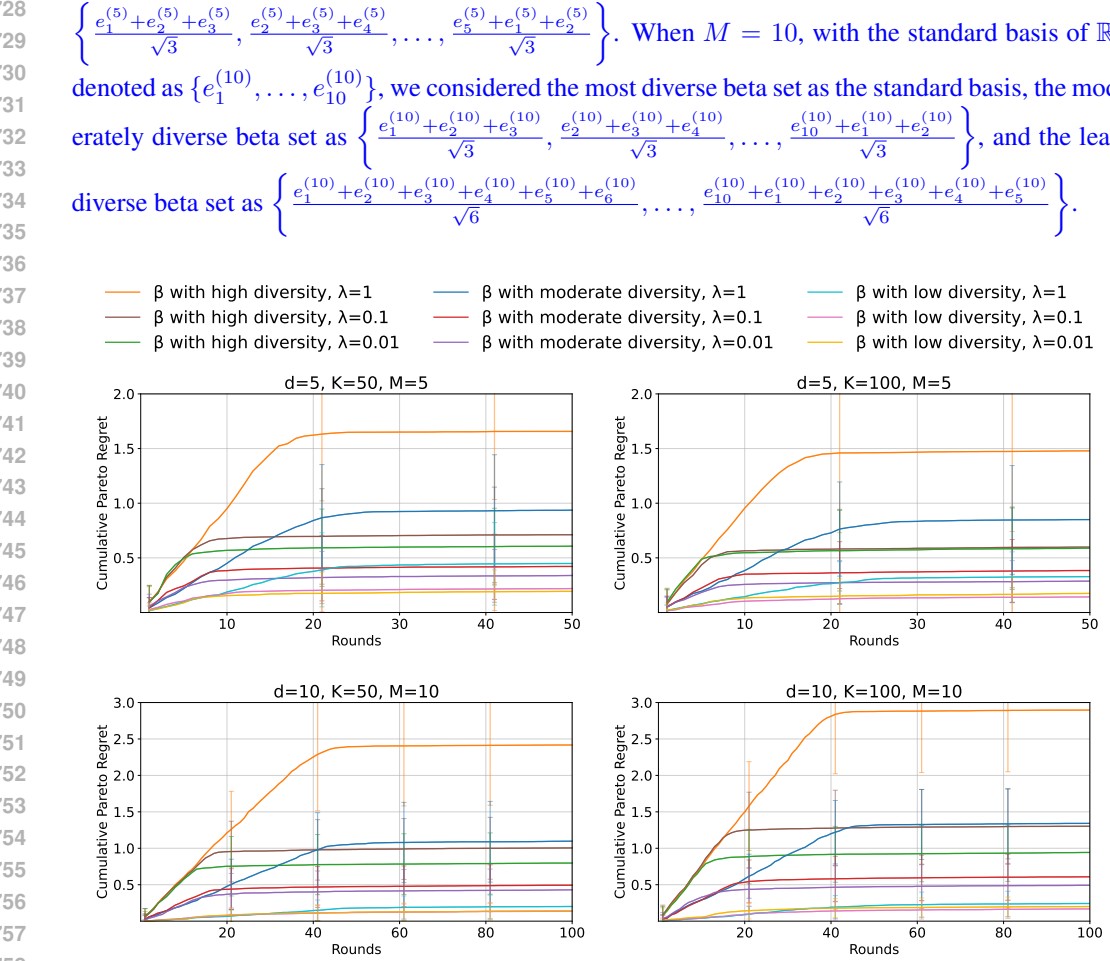

Figure G.5: Performance of `MORR-Greedy` with different initial objective parameters

As shown in Figure G.5 it can be observed that `MORR-Greedy` performs well across all $\{\beta_1, \cdots, \beta_M\}$-sets. First, we note that the performance of the low-diversity $\{\beta_1, \cdots, \beta_M\}$-set is particularly good. This is more related to the distribution of the objective parameters than to the impact of diversity. The low-diversity $\{\beta_1, \cdots, \beta_M\}$-set initializes the objective parameters with vectors located near the center of the positive part of $\mathbb{S}^{d-1}$, and there is a high probability that at least one true objective parameter exists close to these vectors. In contrast, when initializing with a high-diversity $\{\beta_1, \cdots, \beta_M\}$-set, the regret is larger, but it can be observed that exploration terminates earlier in the same $\lambda$ setting. In this case, by selecting diverse features, we can accelerate the increase of the minimum eigenvalue of the Gram matrix. Consequently, we confirmed that our algorithm performs well with initial objective parameters of various diversity levels.

## H  TECHNICAL LEMMAS

**Lemma H.1** (Lemma A.1. of Kannan et al. (2018)). *Let* $\eta_1, \ldots, \eta_t$ *be independent* $\sigma^2$-*subgaussian random variables. Let* $x_1, \ldots, x_t$ *be vectors in* $\mathbb{R}^d$ *with each* $x_s$ *chosen arbitrarily as a function of* $(x_1, \eta_1), \ldots, (x_{s-1}, \eta_{t'-1})$ *subject to* $\|x_s\| \le x_{\max}$. *Then with probability at least* $1 - \delta$,

$$\left\| \sum_{s=1}^{t} \eta_s x(s) \right\| \le \sigma \sqrt{2 x_{\max} d t \log(dt/\delta)}.$$

Note that, the above lemma holds even when $\eta_1, \ldots, \eta_t$ be conditionally $\sigma^2$-subgaussian random variables, because it was driven by using $\sigma^2$-subgaussian martingale.

**Lemma H.2** (Lemma 8 of Li et al. (2017))**.** *Given* $||x_i|| \leq 1$ *for all* $i \in [K]$, *suppose there is an integer* $m$ *such that* $\lambda_{\min}(V_m) \geq 1$, *then for any* $\delta > 0$, *with probability at least* $1 - \delta$, *for all* $t \geq m + 1$,

$$||S_t||^2_{V_t^{-1}} \leq 4\sigma^2(\frac{d}{2}\log(1 + \frac{2t(x_{\max})^2}{d}) + \log(\frac{1}{\delta})).$$

**Lemma H.3** (Theorem 3.1 of Tropp (2011))**.** *Let* $\mathcal{H}_1 \subset \mathcal{H}_2 \cdots$ *be a filtration and consider a finite sequence* $\{X_k\}$ *of positive semi-definite matrices with dimension* $d$ *adapted to this filtration. Suppose that* $\lambda_{\max}(X_k) \leq R$ *almost surely. Define the series* $Y \equiv \sum_k X_k$ *and* $W \equiv \sum_k \mathbb{E}[X_k|\mathcal{H}_{k-1}]$. *Then for all* $\mu \geq 0$, $\gamma \in [0, 1)$ *we have*

$$\mathbb{P}[\lambda_{\min}(Y) \leq (1-\gamma)\mu \ \text{and} \ \lambda_{\min}(W) \geq \mu] \leq d(\frac{e^{-\gamma}}{(1-\gamma)^{1-\gamma}})^{\mu/R}.$$

**Lemma H.4.** *For any random variable vector* $X \sim D$, $\mathbb{E}[XX^\top] \succeq \mathbb{E}[X]\mathbb{E}[X]^\top$

*Proof of Lemma H.4.* For any $u \in \mathbb{S}^{d-1}$, $u^\top \mathbb{E}[XX^\top]u = \mathbb{E}[u^\top XX^\top u] = \mathbb{E}[\langle u, X\rangle^2] \geq (\mathbb{E}[\langle u, X\rangle])^2 = u^\top \mathbb{E}[X]\mathbb{E}[X]^\top u.$

**Lemma H.5.** *Let* $v$ *be a vector in* $S \subset \mathbb{R}^d$ *and* $A$ *be a* $d \times d$ *matrix. Then* $||Av||_2 \geq (\min_{u \in S} u^\top Au) ||v||_2$.

*Proof of Lemma H.5.*

$$\frac{||Av||_2}{||v||_2} = \left|\left|A\frac{v}{||v||_2}\right|\right|_2 \geq \min_{u \in S} ||Au||_2 \geq \min_{u \in S} u^\top Au.$$

