# OpenReview forum: "Mostly Exploration-free Algorithms for Multi-Objective Linear Bandits"
_ICLR.cc/2025/Conference — ICLR 2025 Conference Withdrawn Submission_

### Official Review · Reviewer_1su3 · 2024-10-29

**Soundness:** 2
**Presentation:** 2
**Contribution:** 2
**Rating:** 3
**Confidence:** 4

**Summary:**

This paper investigates multi-objective stochastic linear bandits with finite arms, contributing a novel approach by introducing objective fairness for multi-objective bandits.

**Strengths:**

1. The paper focuses on a central issue in multi-objective optimization.
2. The related work is well-organized.

**Weaknesses:**

1. **Lines 35-40**: The authors frequently mention the increased complexity associated with multi-objective problems but do not specify the complexity level of existing algorithms (Yahyaa and Manderick, 2015; Turgay et al., 2018; Lu et al., 2019; Kim et al., 2023). Could the authors clarify how much the proposed algorithm reduces this complexity? Including a comparison table would be helpful.

2. **Lines 221-233**: The concept of "objective fairness" is unclear. Is it represented by the parameter $p_\epsilon$, or is it a characteristic of a multi-objective bandit algorithm? Please provide a precise definition of objective fairness, similar to how the Pareto order is defined. Additionally, examples illustrating this concept would aid in comprehension. For instance, for two arms with expected rewards $[0,1]$ and $[1,0]$, what kind of decision sequence would qualify as fair?

3. **Line 252**: Are the values $\\{\beta_1, \dots, \beta_M\\}$ initialized according to Definition 4? I could not find the definition of "greedy selection." Please include examples demonstrating the initialization of $\\{\beta_1, \dots, \beta_M\\}$.

4. **Line 314**: The assumption $||\theta_m^*|| = 1$ is not common in single-objective or multi-objective bandit literature, as far as I could ascertain from the cited works (Abbasi-Yadkori et al., 2011; Chu et al., 2011; Agrawal & Goyal, 2013; Abeille & Lazaric, 2017). This assumption appears overly restrictive given that $\theta_m^*$ is unknown.

5. **Line 323**: Is this assumption prevalent in existing studies? Furthermore, why does the assumption that $\theta_1^*, \ldots, \theta_M^*$ span $\mathbb{R}^d$ imply objective diversity? This property seems only to support the objective value of a single fixed arm, while diversity typically refers to variability across different arms.

6. **Line 335**: I could not locate the definition of $\mathbb{B}_\alpha(\cdot)$. Please clarify if I have overlooked it.

7. **Line 375**: What is $T_0$?

8. **Theorem 1**: The term $\delta$ is missing from the regret bound. Perhaps your theorem specifies the expected regret bound $E[PR(T)]$ rather than $PR(T)$.

**Questions:**

Please refer to weaknesses.

---

> ### Author Response · Authors · 2024-11-27
>
> Thank you for taking the time to provide comments on our paper. We sincerely appreciate your thoughtful feedback. We would like to take this opportunity to clarify a few of the points you raised.
>
> **[W1]** Complexity Comparison: In algorithms that construct the Pareto front in each round, the computation time for the empirical Pareto front increases quadratically with the number of arms, which significantly limits scalability as the number of arms grows. Additionally, due to the need for complex vector calculations, these algorithms experience significant slowdowns as the complexity of exploration terms increases. As a result, the algorithm may become impractical to apply in scenarios involving a large number of arms or high-dimensional objectives.
> In contrast, our algorithm simplifies the process by requiring only scalar reward comparisons for the target objective in each round, significantly reducing computational complexity.
>
> **[W2]** Objective Fairness: We have revised the explanation of Objective Fairness in Section 2.2.2 to provide a clearer presentation of this fairness criterion. In essence, Objective Fairness assesses whether an algorithm consistently selects near-optimal arms for each objective, ensuring that no objective is neglected over time.
> For instance, in a multi-objective bandit problem with at least two objectives, applying the UCB algorithm based solely on the first objective can achieve a Pareto regret bound of $\tilde{\mathcal{O}}(d\sqrt{T})$. However, this approach inevitably leads to the exclusion of near-optimal arms for the second objective over time, thereby violating Objective Fairness. In contrast, if both optimal arms are consistently selected across rounds, Objective Fairness is satisfied.
> For a more detailed discussion on Objective Fairness, please refer to Section 2.2.2.
>
> **[W3]** Initialization: First of all, we confirmed that Algorithm 1 performs well when the values of the parameters $\beta_1, \dots, \beta_M$ are sufficiently diverse, both theoretically (Remark 3 and Appendix A.4) and empirically (Appendix G.3). The fundamental principle for initializing $\beta_1, \dots, \beta_M$ is is to ensure the selection of sufficiently diverse arms.
> In the case of fixed arms, once $\beta_1, \dots, \beta_M$ are set, the corresponding arms $z_1, \dots, z_M$ optimized according to $\beta_1, \dots, \beta_M$ will be selected during the exploration process in a round-robin fashion. Therefore, in the fixed arms case, implementing Algorithm 1 only requires selecting $M$ arms $z_1, \dots, z_M$ that span $\mathbb{R}^d$. (Notably, using exploration facilitating initial values is not necessary to obtain the regret bound in Theorem 1.) In practice, when the number of arms $K$ is moderate, one can determine the exploration-facilitating initial values of $\beta$. However, when $K$ is large, the number of possible $\binom{K}{M}$ combinations grows exponentially, making an exhaustive search computationally infeasible. In such cases, methods such as sequentially adding arms (forward selection) or selecting a subset of arms with the largest norms can be employed.
> In scenarios where the arms change over time, initializing $\beta_1, \dots, \beta_M$ with more diverse values—such as ensuring initial selections cover a broad range of possible contexts—facilitates faster adaptation and exploration. In practice, using a basis of $\mathbb{R}^d$ for initialization is effective. If the distribution of contexts is known to some extent, this information can be leveraged to initialize $\beta_1, \dots, \beta_M$ in a way that maximizes the diversity of the selected contexts.
>
> **[W4]** The papers you referenced assume not only that $\theta^*$ is unknown but also that $||\theta^*||$ is bounded. In our study, we restrict the norm to 1 in the main text for the sake of clarity in the analysis. The extension of this assumption to an arbitrary bound is detailed in Appendix E.
>
> **[W5]** To the best of our knowledge, research on the diversity of objectives remains limited. Objective diversity refers to the variation in the objective parameters. Specifically, if the objective parameters span $\mathbb{R}^d$, they cover all directions within $\mathbb{R}^d$, ensuring that the minimum eigenvalue of the Gram matrix formed by these parameters is positive. In this context, we use the term 'diversity' to describe this property.
>
> **[W6]** The definition of $\mathbb{B}_{\alpha}$ is in Section 2.1.
>
> **[W7]** The definition of $T_0$ is in Section 4.1. (Just before Lemma 4.1.)
>
> **[W8]** The definition of Pareto Regert is modified by  $PR(T):=\sum_{t=1}^T \mathbb{E}[\Delta_{a(t)}]$. Thank you for notifying us about this.
>
> Thank you again for your interest and comments! We hope we have addressed your questions and provided the needed clarification in our responses.

---

### Official Review · Reviewer_Y1Nh · 2024-10-31

**Soundness:** 1
**Presentation:** 2
**Contribution:** 1
**Rating:** 3
**Confidence:** 4

**Summary:**

This paper addresses the challenge of multi-objective linear bandits, specifically investigating whether an increasing number of objectives complicates the learning process. To address this question, the authors propose two algorithms that greedily select the optimal arms in each round by alternately dealing with one objective at a time. Theoretical analysis and numerical experiments demonstrate that the algorithms achieve relatively ideal performance.

**Strengths:**

The paper is easy to follow.

**Weaknesses:**

The motivation of the paper needs to be clearer. Can you give more description about the meaning of diverse objectives, which causes the drawbacks of existing methods?

The index of objective fairness is confusing; please provide an intuitive and illustrative explanation to clarify this part. For example, the change of this value in an experiment.

My main concern is the selection strategy for sub-optimal arms. The authors claim that one of the main contributions is the consideration of objective fairness, which involves playing each Pareto optimal arm fairly. However, the algorithms select the arm by alternately maximizing each objective, which only results in selecting special solutions. The authors should refer to conclusions from the multi-objective optimization community. Given this, I highly recommend that the authors investigate the specific needs of this case.

Additionally, the two proposed algorithms differ only in their initial phase, with one using manually set parameters and the other using random parameters. Despite the theoretical analysis provided, I do not see any critical function of this part.

Finally, in the experimental study, the proposed algorithm performed exceptionally well with almost zero regret in all cases, even when the number of arms was up to 100. These results do not seem convincing. More details of the experimental setup should be provided, and using a real-world dataset could offer more powerful verification.

Overall, the quality of the paper needs further improvement.

**Questions:**

See the weakness above.

---

> ### Author Response · Authors · 2024-11-27
>
> Thank you for taking the time to provide comments on our paper. We sincerely appreciate your thoughtful feedback. We would like to take this opportunity to clarify a few of the points you raised.
>
>
> **[W1]** Objective diversity: First of all, the term "objective diversity" refers to the diversity of the objective parameters. For example, in a situation where both the dimension of the feature space and the number of objectives are 2, if the two objective parameters $\theta_{1}^*$ and $\theta_{2}^*$ span $\mathbb{R}^2$, then objective diversity is achieved.
> Importantly, objective diversity does not lead to the drawbacks associated with existing methods. Our argument is that "there exist algorithms that are faster and more efficient than existing ones when objective diversity is taken into account."
>
> **[W2]** Objective Fairness: We have updated the explanation of 'Objective Fairness' in Section 2.2.2 to clearly present our idea of this new fairness criterion. Objective Fairness is a criterion for multi-objective bandit algorithms that examines whether an algorithm consistently considers all optimal arms for each objective. In other words, the algorithm should ensure that near-optimal arms are consistently selected for each objective, preventing any objective from being disregarded over time. It can serve as one of the goals of a multi-objective bandit algorithm, alongside Pareto regret minimization.
> The objective fairness index refers to the proportion of rounds in which $\epsilon$-optimal arms are selected for the least chosen objective. However, this index is not a parameter that can be directly manipulated or controlled. Instead, we conducted experiments to empirically estimate the objective fairness index of our algorithm; detailed results can be found in Appendix G.2.
>
> **[W3]** Selection Strategy: First of all, our algorithm does not employ a strategy of selecting sub-optimal arms. On the contrary, it aims to maximize exploitation as much as possible. The key distinction is that **Objective fairness is not about whether the Pareto optimal arm is selected fairly.** The fairness you refer to is the criterion proposed by [Drugan & Nowe](https://ieeexplore.ieee.org/document/6707036) , which we refer to as Pareto front fairness, and it differs from our concept of fairness.
> Before elaborating on our approach, we would like to emphasize that Pareto regret minimization and Pareto front approximation are distinct goals of multi objective bandit algorithms. As noted in [Xu and Klabjan](https://arxiv.org/abs/2212.00884), focusing solely on Pareto regret minimization allows algorithms to optimize for a specific objective, yielding regret bounds comparable to those in single-objective settings. Most of existing algorithms, such as [P-UCB](https://ieeexplore.ieee.org/document/6707036) and [MOGLM-UCB](https://arxiv.org/abs/1905.12879), targeting both Pareto regret minimization and Pareto front fairness typically select points from the Pareto front with equal probability.
> Our algorithm, however, targets Objective Fairness, which may be more beneficial in some situations. For instance, there might be situations that users want to ensure that at least one of their objectives achieves the optimal reward. Additionally, if estimating the Pareto front is not necessary, our algorithm, which achieves Objective Fairness while ensuring fast execution, may be a more suitable choice.
>
> **[W4]** Two proposed algorithms: As you pointed out, the two proposed algorithms share significant similarities. However, we present both because MORR-Greedy is a deterministic algorithm, while MORO-Greedy incorporates randomness in its selection process. We aim to demonstrate that our approach remains effective even when stochastic elements are introduced. Additionally, the two algorithms differ not only in the initial phase but also in the way they select the target objective in each round.
>
> **[W5]** Experiment: First of all, achieving zero regret is not an unusual outcome in our problem setting. Our study specifically addresses cases where the number of objectives exceeds the feature dimension. As the number of objectives increases, the Pareto front includes more arms, and by the definition of Pareto regret, selecting any arm from this front results in zero regret. Additionally, because the greedy selection strategy quickly converges to the optimal objective parameters, it allows for a faster distinction between optimal and sub-optimal arms.
> Moreover, conducting experiments with real datasets in bandit algorithm studies is challenging, given that selections and rewards are obtained sequentially. As a result, many studies, including ours, rely on simulated experiments instead.
>
> Thank you again for your interest and comments! We hope we have addressed your questions and provided the necessary clarification in our responses.

---

### Official Review · Reviewer_Ndmz · 2024-11-01

**Soundness:** 2
**Presentation:** 3
**Contribution:** 2
**Rating:** 3
**Confidence:** 3

**Summary:**

This paper tackles solving multi-objective linear bandit problems, where multiple, potentially conflicting objectives need to be optimized simultaneously. The authors propose to leverage the diversity among objectives to drive exploration, instead of relying on the diversity of the feature vectors (contexts) as traditionally done.  They introduce two near-greedy algorithms, MORR-Greedy (Multi-Objective Round Robin Greedy) and MORO-Greedy (Multi-Objective Random Objective Greedy), which are simpler and more computationally efficient than existing methods because they don't require constructing the empirical Pareto front in each round.

Under the assumptions of objective diversity (objective parameters spanning the feature space) and a newly introduced regularity condition (γ-regularity), the authors prove that their algorithms achieve a regret bound of sqrt(dT). They also introduce a concept called "objective fairness," ensuring that all objectives are considered equitably, and demonstrate that their algorithms satisfy this criterion.

**Strengths:**

Originality: The paper presents a novel perspective on multi-objective bandit problems by highlighting the potential of objective diversity to induce free exploration.

Quality:  The paper mainly provides a theoretical analysis of how objective diversity aids the search, establishing regret bounds of  sqrt(dT) for the proposed MORR-Greedy and MORO-Greedy algorithms under the objective diversity and γ-regularity assumptions. The introduction and analysis of the objective fairness criterion further is interesting.

Clarity: The paper is generally well-written and clearly presents the problem setting, algorithms, and main results.

Significance: If the theoretical results hold and generalize well, the paper's findings could have significant practical implications.

**Weaknesses:**

Major Soundness Issue: Under your setting, there is a stochastic linear bandit lower bound of d*sqrt(T) [see https://tor-lattimore.com/downloads/book/book.pdf], although for adversarial linear bandits, there is an upper bound of sqrt(dT) only for the action set being the L2 ball [see https://arxiv.org/pdf/1711.01037 {Sparsity, variance and curvature in multi-armed bandits}: it shows that for the $l_p$ norm ball linear bandit problem($p>2$), no algorithm could have better that Omega(d sqrt(T))].

Since your results should extend to the single-objective case, Pareto regret becomes regular cumulative regret and your assumptions on  γ-regularity still hold and K can be exponentially large. Then, it appears that your upper bound is too good to be true, can you explain this?

Connection to Pareto Front Approximation: The paper focuses on minimizing Pareto regret, but doesn't explicitly discuss how the proposed algorithms relate to the goal of approximating the Pareto front. Specifically, the fairness objective tries to address this by  inducing approximation to the maxima of the M objectives separately, it still does not approximate the Pareto front as there are multiple directions in the "middle" of the Pareto front that needs to be approximated. Adding this as a discussion is important and a recent paper (https://arxiv.org/abs/2307.03288) has shown that in general, there is in fact a T^{-1/M} lower bound. In light of the scalarizations framework of that paper, it appears that you are simply considering closeness in only M directions and it is unclear why this would lead to a good multiobjective search. As proposed in that paper, would it make more sense to have multiple random scalarization directions?


Limited Empirical Validation:  While the paper includes some numerical experiments, they are relatively limited in scope. They do not involve the fairness objectives that were introduced and it would be also interesting to include hypervolume regret as a more robust version of multiobjective progress. It would also be interesting to see experiments with varying degrees of objective diversity and γ-regularity to understand how these factors impact algorithm performance.

Clarity on γ-Regularity: While the paper introduces the concept of γ-regularity, its practical implications and relationship to real-world datasets are not fully clear. Specifically, it is used to implicitly assume a feature diversity in terms of the isotropic nature of the gram matrix and provide a connection between objective and feature diversity. However, the analysis still hinges on the underlying feature diversity, making the messaging on the paper extremely confusing. More intuitive explanations and examples of datasets that exhibit (or do not exhibit) γ-regularity would be helpful. It would also be beneficial to discuss methods for estimating γ and α0 from data.

**Questions:**

Generally, no algorithm could have better that Omega(d sqrt(T)) regret for singleobjective optimization with K = exponentially many arms. Can you explain how your upper bound seems too good to be true?

Can you explain why you claim that "we do not assume any such diversity in the features", given that γ-regularity assumption connects objective with feature diversity?

In light of https://arxiv.org/abs/2307.03288, why is the fairness criterion good for multiobjective optimization and should fairness be defined with respect to more directions than just the maxima of M objectives?

---

> ### Author Response · Authors · 2024-11-27
>
> We are grateful for the depth of insight demonstrated in your review, and we greatly appreciate the important questions you raised. As you mentioned, our research centers on leveraging objective diversity as a mechanism to induce free exploration in multi-objective bandit problems. We would like to take this opportunity to clarify several key points to ensure a deeper understanding of our contributions.
>
> **1. Soundness**
> We are well aware that no stochastic linear bandit algorithm can achieve a regret bound better than $O(d\sqrt{T})$.
> However, it is important to note that the regret bound we obtained in Theorem 1 does not contradict this.
> The reason is that the environmental parameters in our problem setting, such as $\lambda_0$, $\gamma_0$,and $\alpha_0$, may potentially include terms like $d, M$ and $K$. This phenomenon, where the environmental parameters in the assumptions reduce the order of $d$ and $T$, is commonly observed in the literature on free exploration, such as [Kannan et al.](https://arxiv.org/abs/1801.03423) and [Bastani et al.](https://arxiv.org/abs/1704.09011). (We have revised Section 4 of our paper to provide a clearer explanation of the regret bound.)
>
> **2. Pareto Front Approximation**
> First, we would like to emphasize that Pareto regret minimization and Pareto front approximation are **distinct goals**. As noted in [Xu and Klabjan](https://arxiv.org/abs/2212.00884), focusing solely on Pareto regret minimization allows algorithms to optimize for a particular objective, yielding regret bounds comparable to those in single-objective settings. Therefore, meaningful multi-objective bandit studies often pursue additional goals, typically fairness, alongside Pareto regret minimization.
>
> Algorithms targeting both Pareto regret minimization and Pareto front approximation typically select points from the middle of the Pareto front with equal probability. In other words, these algorithms fulfill the fairness criterion proposed by [Drugan & Nowe](https://ieeexplore.ieee.org/document/6707036). Our algorithm, in contrast, aims for both Pareto regret minimization and Objective Fairness. (For details, please refer to Section 2.2.2).
>
> In some scenarios, objective fairness may be more advantageous. For instance, users may want to ensure that at least one of their objectives achieves the optimal reward. Additionally, there may be situations where not all Pareto optimal points are relevant or of interest. Therefore, in some situations, algorithms that satisfy Objective Fairness with faster execution times may be more practical and beneficial compared to those focused on approximating the Pareto front.
>
> **3. Empirical validation**
> We conducted experiments to empirically estimate the objective fairness index of our algorithm and confirmed that it consistently selects near-optimal arms for each objective. (see Appendix G.2.) Moreover, we confirmed that our algorithm performs consistently well across various levels of diversity and regularity, with no significant performance changes observed. Additionally, we found that our algorithm performs robustly across various levels of diversity in initial parameters. (see Appendix G.3)
> We appreciate the concept of hyper volume regret that you mentioned, which offers a valuable perspective. We will consider further analyzing our algorithm using this regret measure.
>
> **4. $\gamma$-regularity and feature diversity**
> As you noted, $\gamma$-regularity and objective diversity do lead to feature diversity. However, we clarify that the diversity we refer to here differs from the context diversity assumed in previous studies on free exploration, which typically focuses on stochastic contexts and their randomness. For example, in one of the most representative studies on free exploration, [Bastani et al.](https://arxiv.org/abs/1704.09011) assume the existence of $\lambda_0$ such that for each vector $u \in \mathbb{R}^d$, $\lambda_{\min}(\mathbb{E}_X[XX^\top \mathbb{I} \{X^\top u \ge 0\} ]) \ge \lambda_0$.
>
> In contrast, the $\gamma$-regularity assumption in our setting addresses the distribution of feature vectors along the direction of the objective parameters. Combined with objective diversity, this leads to feature diversity that is independent of context randomness and thus applies even in the fixed-arm case.
>
> Intuitively, a dataset exhibiting $\gamma$-regularity should allow contexts to cover all directions in the feature space. It is not necessary for each arm to cover the entire  feature space; instead, it is sufficient to partition the feature space into regions, with each arm covering a specific region. Further details about $\gamma$-regularity can be found in Appendix C.
>
> Thank you again for your interest and comments! We hope we have addressed your questions and provided the needed clarification.

---

> > ### Comment · Reviewer_Ndmz · 2024-12-02
> >
> > For 1, which is my major concern, I'm not sure if you addressed my concerns or clearly shown how your additional parameters would satisfy the lower bound. Previous results in the contextual linear bandit setting are not clearly relevant, at least not to me.
> >
> > For 2, it appears that objective fairness is simply a way to "get to" Pareto front approximation. I still find the objective fairness direction a bit limiting in terms of applications as there was no clear problems that would require this.
> >
> > For 3, just as an aside, I would make this into the main paper, as the fairness appears to be a main contribution.
> >
> > For 4, you mentioned that "\gamma-regularity assumption in our setting addresses the distribution of feature vectors", so you are making feature diversity assumptions. You may be overloading the word feature, as you often cite previous works in contextual linear bandit setting, whereas here its a fixed hidden parameter. This is not a huge problem per say but it really makes things confusing.
> >
> > Given the above, I've decided to keep my score.

---

> ### Author Response · Authors · 2024-12-03
>
> Thank you for your valuable feedback.
>
> - **For Issue 1**, we would like to clarify this point as it is a critical part of our analysis. We aim to provide a concrete example where the order of $d$ and $T$ is reduced due to the environmental parameters used in our assumptions. For instance, in [Kannan et al.](https://arxiv.org/pdf/1801.03423), a final bound of $O(\sqrt{Td}/\sigma^2)$ was derived, where the dependence on $d$ is lower compared to the lower bound of $d\sqrt{T}$. (Here, $\sigma$ is a parameter included in the assumption related to context perturbation.) Similarly, the regret bound of $O(\sqrt{dT}/\lambda_0)$ that we propose in our study also exhibits the same phenomenon. As mentioned earlier, this is because $\lambda_0$ may implicitly include variables such as $d$ and $M$. We plan to explore the relationship between $\lambda_0$ and $d$ in future work to provide a clearer and sufficient explanation of the lower bound issue.
>
> - **For Issue 2**, we will propose more specific problem settings where an algorithm satisfying objective fairness is necessary or where considering only optimal points in certain directions is sufficient.
> - **For Issues 3 and 4**, we will address these by improving the clarity of our presentation, and we greatly appreciate your insights in pointing out these areas.
>
> Your comments have provided valuable input for refining our analysis and presentation, and we are confident they will contribute to making our work clearer and more robust. Thank you once again for your constructive feedback.

---

### Official Review · Reviewer_6Rxe · 2024-11-05

**Soundness:** 2
**Presentation:** 3
**Contribution:** 2
**Rating:** 6
**Confidence:** 3

**Summary:**

This paper considers multi-objective linear bandit problems. Unlike existing complex methods, this study leverages "objective diversity" to enable free exploration, leading to efficient algorithms (MORR-Greedy and MORO-Greedy) that achieve good regret bounds without constructing empirical Pareto fronts.

**Strengths:**

1. The paper is clearly written.
2. The idea of "objective diversity induces free exploration" is novel.
3. The proposed algorithm is simple and effective.
4. The concept of objective fairness is another contribution, ensuring equitable treatment across all objectives.

**Weaknesses:**

1. The introduction could more directly highlight practical scenarios where current multi-objective algorithms face scalability or computational limitations due to their reliance on Pareto front constructions and complex exploration mechanisms. Illustrating these challenges would provide a compelling reason for why a simpler, exploration-free method like MORR-Greedy is beneficial.
2. The proposed algorithms assume objective diversity to enable free exploration. Whether this diversity exists naturally in most multi-objective setups is not clear. In real-world scenarios, it is likely that objectives are correlated. Could you provide examples or discuss scenarios where objective diversity is likely to occur naturally, or address how your methods might perform when objectives are correlated?
2. The numerical experiments, while supportive of the theoretical claims, may be overly simplified. How the hyper-parameters are chosen is also not discussed.

**Questions:**

1. Why $\gamma$-regularity is a proper assumption? How likely it can happen in reality? Could you provide intuition or examples of when $\gamma$-regularity naturally occur in practical scenarios?
2. How do you show $T_0$ is independent of $T$? Since $T$ shows up in the expression of $\lambda$, I think it is necessary to show $T_0$ is of order $O(\sqrt{T})$. It seems that in the paper it is simply treated as a constant.
3. Could you elaborate more on your claim that "objective diversity induces free exploration"? I am not clear on where this concept is applied within the proof.

---

> ### Author Response · Authors · 2024-11-27
>
> We sincerely appreciate your interest and the thoughtful comments you provided. As you pointed out, our research focuses on developing simple and efficient algorithms aimed at achieving optimal regret and free exploration. This approach is motivated by the inherent diversity of objectives present in multi-objective bandit problems. We would also like to take this opportunity to clarify a few key points, ensuring a comprehensive understanding of our contributions.
>
>
> **[W1]**. In algorithms that construct the Pareto front in each round, the computation time for the empirical Pareto front increases quadratically with the number of arms, which significantly limits scalability as the number of arms grows. Additionally, due to the need for complex vector calculations, these algorithms experience significant slowdowns as the number of arms and objectives increases. Furthermore, the algorithm may become impractical to apply in scenarios involving a large number of arms or high-dimensional objectives.
> In contrast, our algorithm simplifies the process by requiring only scalar reward comparisons for the target objective in each round, significantly reducing computational complexity. As a result, our algorithm offers a substantial speed advantage over algorithms that rely on constructing the empirical Pareto front, particularly when dealing with a large number of arms and objectives.
>
> **[W2]**. First, the "objective diversity" we refer to concerns the diversity of the objective parameters. In a typical multi-objective bandit problem, the objective parameters $\theta_1^*, \dots, \theta_M^*$ are assumed to be unknown fixed values rather than random variables, making the concept of correlation between them is generally inapplicable. If by "correlated objectives" you are referring to the observed reward values for each objective being correlated, note that such correlation does not necessarily imply a lack of objective diversity. For example, consider a scenario where the feature space has two dimensions and the number of objectives is also two. If the two objective parameters $\theta_1^*$ and $\theta_2^*$ span $\mathbb{R}^2$, objective diversity is still satisfied.
>
> **[W3]** Details of the experimental design, including the setup, parameter choices, and evaluation metrics, can be found in Appendix G. For example, it describes how the true objective parameters and arms were systematically generated for each repetition, as well as how the performance varied with different tuning parameters for each algorithm to identify optimal settings for each case. Additionally, we conducted experiments to empirically evaluate the objective fairness index of our algorithm, and assessed its performance under various initial objective parameter settings. The results, which illustrate the effectiveness of different parameter settings and fairness metrics, are provided in Appendix G.2 and G.3.
>
>
> **[Q1]** $\gamma$-regularity refers to the condition where, for each objective parameter, there is at least one arm that is sufficiently close to the direction of the objective parameter (in the fixed arms case), or the probability of such an arm existing exceeds a defined threshold (in the stochastic context case). Intuitively, a dataset exhibiting $\gamma$-regularity ensures that contexts are able to cover all directions within the feature space. It is not required for each arm to cover the entire surface of the feature space; rather, it is acceptable for the surface to be partitioned into regions, with each arm covering a specific region. Further details are provided in Appendix C.
>
> **[Q2]**  As you mentioned, $T_0$ depends on $\lambda$ and is bounded by $\tilde{\mathcal{O}}(\min(d \log T, \sqrt{dT}))$, which is discussed in detail in Remark 3 and Appendix A.4 of our paper.
>
> **[Q3]** To summarize, the MORR-Greedy algorithm iteratively selects the optimal arm for each objective. When the objectives are sufficiently diverse, the arms selected by each objective will naturally reflect this diversity. Mathematically, this diversity ensures a constant lower bound for the minimum eigenvalue of the Gram matrix formed by the arms selected in each round. This concept is discussed in Section 3.2, and the proof can be found in Appendix A.1.
>
> Thank you again for your valuable feedback. We hope our responses have addressed your questions comprehensively and clarified any uncertainties.

---

> > ### Comment · Reviewer_6Rxe · 2024-11-27
> > **Thank you for feedback**
> >
> > I think most of my concerns have been addressed. I will raise my score.

---

### Official Review · Reviewer_89dX · 2024-11-06

**Soundness:** 3
**Presentation:** 2
**Contribution:** 2
**Rating:** 5
**Confidence:** 3

**Summary:**

This paper addresses the challenge of balancing multiple objectives in linear bandit problems, proposing a novel approach where objective diversity naturally induces "free exploration," allowing simpler algorithms to perform efficiently. Instead of relying on constructing Pareto fronts, which can be computationally intensive, the paper introduces two algorithms, MORR-Greedy and MORO-Greedy, which use round-robin and near-greedy selections to optimize multiple objectives without requiring explicit exploration. The algorithms achieve a regret bound of $\tilde{O}(dT)$, leveraging objective diversity to enhance performance and reduce the need for complex exploratory steps. This paper also introduces the concept of "objective fairness," ensuring equitable treatment across objectives.

**Strengths:**

By using objective diversity for free exploration, the proposed approach presents a new perspective on how multiple objectives can simplify rather than complicate the decision-making process.

**Weaknesses:**

1. It is hard for me to understand the concept of objective fairness as defined in Section 2.2.2. Is the equation in this section a correct definition of objective fairness? For example, could you provide an intuitive explanation of what the equation in Section 2.2.2 means in practical terms or in real-world applications. Particularly, could you clarify if this definition allows for objectives of different importance, and if so, how that would be incorporated and affect the final result? A clear explanation about these would be easier for me to understand the proposed results.

2. The parameter initialization for $\beta$ in Algorithm 1 is unclear to me. For example, could you provide specific guidelines or a step-by-step procedure for choosing the initial $\beta$ values. Also, the intuition behind Definition 4 and how it relates to algorithm performance are unclear to me. It would be better if a small case study or numerical examples showing how different initializations of $\beta$ might impact the algorithm's behavior and final performance can be provided.

**Questions:**

See weaknesses

---

> ### Author Response · Authors · 2024-11-27
>
> We sincerely appreciate your interest and the thoughtful comments you provided. As you pointed out, our research focuses on developing simple and efficient algorithms aimed at achieving optimal regret and free exploration. We would also like to take this opportunity to clarify a few key points, ensuring a comprehensive understanding of our contributions.
>
> **[W1] Objective fairness** :  We have revised the explanation of 'Objective Fairness' in Section 2.2.2 to present our concept of this new fairness criterion with greater clarity and precision. For a detailed discussion on objective fairness, please refer to Section 2.2.2.
>
> In multi-objective bandit algorithms, fairness is a crucial goal alongside Pareto regret minimization, emphasizing how impartially the algorithm addresses multiple equivalent objectives. We introduce a novel perspective: Objective Fairness, which ensures that an algorithm takes into account all optimal arms for each individual objective. Specifically, the algorithm should consistently select near-optimal arms for each objective, ensuring that no objective is neglected over time.
>
> For instance, in a multi-objective bandit problem with two objectives, using the UCB algorithm solely for the first objective can achieve a Pareto regret bound of $\tilde{\mathcal{O}}(d\sqrt{T})$. However, this approach inevitably neglects near-optimal arms for the second objective, thereby failing to meet the criterion of objective fairness.
>
> There are real-world scenarios where objective fairness is crucial in the application of multi-objective bandit algorithms. For instance, users may wish to ensure that at least one of their objectives consistently achieves an optimal reward.
>
> Notably, objective fairness remains achievable even when objectives vary in importance, as it only requires a common lower bound on the proportion of times near-optimal arms are selected for each objective, without enforcing equal proportions. For example, the MORO algorithm we propose in Appendix D, despite assigning different weights to each objective, still meets the criterion of objective fairness.
>
> In conclusion, our algorithm demonstrates a dual achievement: attaining optimal Pareto regret bounds while ensuring consistent selection of optimal arms for all objectives over time, thereby advancing both efficiency and fairness.
>
> **[W2]  Parameter initialization** : The values of the parameters $\beta_1, \dots, \beta_M$ in Algorithm 1 determine the number of exploration rounds, $T_0$, since they directly influence the rate at which the minimum eigenvalue of the Gram matrix increases throughout the exploration process. Therefore, the key principle for initializing $\beta_1, \dots, \beta_M$ is to ensure that sufficiently diverse arms are selected.
>
> In the case of fixed arms, once $\beta_1, \dots, \beta_M$ are set, the arms $z_1, \dots, z_M$ optimized to $\beta_1, \dots, \beta_M$ will be selected during the exploration phase in a round-robin fashion. In the fixed arms case, implementing Algorithm 1 only requires selecting $M$ arms $z_1, \dots, z_M$ that span $\mathbb{R}^d$ (see Remark 3). If the arms $z_1, \dots, z_M$ are chosen to maximize the minimum eigenvalue of the Gram matrix, this ensures the initial exploration process is as efficient as possible by promoting a well-distributed exploration of the space. This is the essence of exploration facilitating, as defined in Definition 4. (Notably, using exploration facilitating initial values is not necessary to obtain the regret bound in Theorem 1.) In practice, when the number of arms $K$ is moderate, one can determine the exploration-facilitating initial values of $\beta$. However, when $K$ is large, the number of possible $\binom{K}{M}$ combinations grows exponentially, making an exhaustive search computationally infeasible. In such cases, methods such as sequentially adding arms (forward selection) or selecting a subset of arms with the largest norms can be employed.
>
> In scenarios where the arms change over time, initializing $\beta_1, \dots, \beta_M$ with more diverse values—such as ensuring initial selections cover a broad range of possible contexts—facilitates faster adaptation and exploration. In practice, using a basis of $\mathbb{R}^d$ for initialization is effective. If the distribution of contexts is known to some extent, this information can be leveraged to initialize $\beta_1, \dots, \beta_M$ in a way that maximizes the diversity of the selected contexts.
>
> The results of our experiments on objective parameter initialization, detailed in Appendix G.3, demonstrate that our algorithm performs robustly across various levels of diversity in initial parameters.
>
> Thank you again for your interest and comments! We hope we have addressed your questions and provided the needed clarification.

---

### Note · Authors · 2024-12-03

**Comment:**

We have decided to withdraw our submission to further improve our research and ensure a more robust analysis. We sincerely appreciate the interest and support shown for our work.

**Withdrawal Confirmation:**

I have read and agree with the venue's withdrawal policy on behalf of myself and my co-authors.